# Synaptic connectome of the *Drosophila* circadian clock

Nils Reinhard [1,4], Ayumi Fukuda[2,4], Giulia Manoli [1], Emilia Derksen[1], Aika Saito[2], Gabriel Möller [1], Manabu Sekiguchi [2], Dirk Rieger [1], Charlotte Helfrich-Förster [1]✉, Taishi Yoshii [2] & Meet Zandawala [1,3]✉

The circadian clock and its output pathways play a pivotal role in optimizing daily processes. To obtain insights into how diverse rhythmic physiology and behaviors are orchestrated, we have generated a comprehensive connectivity map of an animal circadian clock using the *Drosophila* FlyWire brain connectome. Intriguingly, we identified additional dorsal clock neurons, thus showing that the *Drosophila* circadian network contains ~240 instead of 150 neurons. We revealed extensive contralateral synaptic connectivity within the network and discovered novel indirect light input pathways to the clock neurons. We also elucidated pathways via which the clock modulates descending neurons that are known to regulate feeding and reproductive behaviors. Interestingly, we observed sparse monosynaptic connectivity between clock neurons and downstream higher-order brain centers and neurosecretory cells known to regulate behavior and physiology. Therefore, we integrated single-cell transcriptomics and receptor mapping to decipher putative paracrine peptidergic signaling by clock neurons. Our analyses identified additional novel neuropeptides expressed in clock neurons and suggest that peptidergic signaling significantly enriches interconnectivity within the clock network.

Almost all living organisms from humans to bacteria possess a circadian clock[1]. This internal timekeeping system enables organisms to anticipate and adapt to the rhythmic environmental changes that occur over a 24-hour cycle. At their molecular core, these clocks are comprised of cell-autonomous transcription-translation negative feedback loops[2]. In most animals, the master circadian clock in the brain receives light cues via the eyes which enable synchronization (or entrainment) with the external 24-hour light-dark cycles. The master clock sits at the top of the hierarchy and in turn modulates the activity of downstream neurons, as well as the peripheral clocks located in tissues throughout the body via endocrine and systemic signaling. In vertebrates, the master clock is located in the

suprachiasmatic nucleus (SCN) of the hypothalamus and is comprised of approximately 20,000 neurons[3]. Extensive intercellular coupling between these neurons likely via neurotransmitters, neuropeptides, and gap junctions forms a neuronal network that is resilient to internal and environmental perturbations[4,5]. Systematic characterization of these diverse coupling mechanisms between clock neurons is thus crucial to understanding circadian clock entrainment and mutual coupling of the clock neurons. In addition, unraveling the clock output pathways which generate rhythmic behaviors and hormonal signaling can provide mechanistic insights into circadian regulation of organismal physiology and homeostasis.

[1]Neurobiology and Genetics, Theodor-Boveri-Institute, Biocenter, Julius-Maximilians-University of Würzburg, Am Hubland, Würzburg, Germany. [2]Graduate School of Natural Science and Technology, Okayama University, Okayama, Japan. [3]Department of Biochemistry and Molecular Biology and Integrative Neuroscience Program, University of Nevada Reno, Reno, NV, USA. [4]These authors contributed equally: Nils Reinhard, Ayumi Fukuda. ✉e-mail: charlotte.foerster@uni-wuerzburg.de; mzandawala@unr.edu

These aims are particularly challenging to accomplish in vertebrates due to the large neuronal network size and resultant increase in complexity. However, the molecular clock architecture as well as the neuronal network motifs are highly conserved from humans to insects[6,7]. Hence, *Drosophila melanogaster* with its powerful genetic toolkit and a complete brain connectome represents an ideal system for deciphering the clock network and its input and output pathways[8–10]. Not surprisingly, the fly circadian clock, generally regarded to be comprised of approximately 150 neurons, is extremely well characterized[11]. These neurons have historically been classified into different groups of Lateral and Dorsal Neurons (LN and DN, respectively) based on their size, anatomy, location in the brain, and differences in gene expression[12]. Further, these neuronal subgroups are active at different times during the day and are consequently distinct functionally[13]. Single-cell transcriptome sequencing analyses of clock neurons recently revealed additional heterogeneity within some of these clusters, which can largely be explained by neuronal signaling molecules that they express[14,15]. Comparing the synaptic connectivity of different clock neurons can determine if the molecular heterogeneity based on gene expression also translates into heterogenous synaptic connectivity. Nonetheless, given the rich array of neuropeptides expressed in clock neurons, both synaptic and paracrine signaling appear crucial in mediating the connectivity between clock neurons as well as their output pathways. While recent work has begun to uncover some of this connectivity[16–21], global analyses encompassing entire neuronal networks across both brain hemispheres are lacking.

Here, we harnessed the power of connectomics to generate the first comprehensive connectivity map of the circadian clock. Intriguingly, we identified additional DN in the network, thus showing that the *Drosophila* circadian clock is in fact comprised of at least 240 neurons instead of 150 neurons. In addition, our analyses revealed that light input from extrinsic photoreceptors to the clock neurons is largely indirect. Furthermore, we discovered extensive ipsilateral synaptic connectivity between the clock neurons and identified a subset of DN as an important hub that link the clock network across the two brain hemispheres via contralateral projections. We also elucidated the output pathways from the clock network that could affect general behavioral activity levels and organismal physiology. In particular, we elucidated clock output pathways to descending neurons that are known to regulate feeding and reproductive behaviors. Additionally, we observed sparse monosynaptic connectivity between clock neurons and higher-order brain centers, suggesting that multi-synaptic connections and peptidergic signaling account for most of the connectivity, as is characteristic of the vertebrate clock output pathways[22]. Hence, as a complementary approach, we also deciphered putative paracrine signaling pathways within the clock network by extensively mapping the expression of clock neuropeptides and their receptors, and filtering them based on the topographical constraints determined by the connectome. This peptidergic signaling greatly enriches the connectivity within the clock network and appears to be the major output pathway to the neuroendocrine center[23,24] which regulates systemic physiology.

## Results

### Identification of the circadian neuronal network in the FlyWire and hemibrain connectomes

The master clock in the *Drosophila* brain is widely regarded to be comprised of approximately 150 neurons. This number is derived from neuronal expression of different clock genes[14]. *Clk856-Gal4*, based on the *Clk* promoter, faithfully recapitulates expression in most of these clock neurons (Fig. 1A). These neurons can be broadly classified into four classes each of LN and DN (Fig. 1B and Table 1). The LN comprise Lateral Posterior Neurons (LPN), dorsoLateral Neurons ($LN_d$), Ion Transport Peptide (ITP)-expressing LN ($LN^{ITP}$), and Pigment-Dispersing

Factor (PDF)-expressing ventroLateral Neurons ($LN_v^{PDF}$). Conversely, the DN include anterior Dorsal Neurons 1 ($DN_{1a}$), posterior $DN_1$ ($DN_{1p}$), $DN_2$ and $DN_3$. These clock neuron classes can be further subdivided into different cell types (Fig. 1C and Table 1). Morphological characterization of some $DN_{1p}$ and $DN_3$ subtypes is currently lacking[18]. Moreover, the estimated number of clock neurons in *Drosophila* is likely an underestimation as the precise number of $DN_3$ has not been determined thus far[25]. This is partly because drivers like *Clk856-Gal4* only include a small proportion of $DN_3$[14]. In addition, most $DN_3$ have small somata that are densely packed together, making it difficult to count. The majority of these small $DN_3$ project to the central brain, and hence they are aptly called small Central Projecting $DN_3$ (s-$CPDN_3$) (Table 1)[26]. Similarly, a pair of $DN_3$ with large somata project to the central brain (large Central Projecting $DN_3$; l-$CPDN_3$), whereas about 6 neurons per brain hemisphere project to the anterior brain (Anterior Projecting $DN_3$; $APDN_3$)[26]. These latter cells also have larger somata than the s-$CPDN_3$[18]. Despite the large number of $DN_3$, previous studies suggest that they are less important for behavioral rhythmicity compared to some of the LN, which are regarded as the master pacemaker neurons[27–29].

As a first step in determining the synaptic connectivity within the clock network, we used a systematic approach (Supplementary Fig.1) to identify most, if not all, of the clock neurons in the FlyWire connectome generated in our companion papers[9,10]. In total, we successfully identified 242 clock neurons based on a combination of morphology, previously determined connectivity, and location of their cell soma (Fig. 1C, D, Supplementary Movies 1, 2, Supplementary Data 1)[17–19,26,30]. The number of clock neurons that we identified in the connectome was considerably higher than the expected number of clock neurons (~152) in the adult brain (Table 1). This was mainly due to the presence of more s-$CPDN_3$ in the connectome than estimated, which prompted us to accurately quantify the total number of $DN_3$ in adult brains. To comprehensively quantify $DN_3$, we used antibodies against clock proteins Period (PER) and Vrille (VRI) in *tim-Gal4 > GFP* expressing flies. Since PER and VRI exhibit peak expression at different Zeitgeber times (Fig. 2A), the inclusion of three clock markers (PER, VRI, and *timeless* (*tim*)) can provide a reliable estimate of the total number of $DN_3$ (Fig. 2B, C, Supplementary Movie 3). Indeed, our analyses revealed that there are more than 70 $DN_3$ per hemisphere in both males and females (Fig. 2D). Thus, the number of $DN_3$ identified in the connectome (171 neurons) is in line with our $DN_3$ estimate (~166 neurons) based on anatomical analysis. The majority of these are s-$CPDN_3$ (157 neurons) which we now classified into five subtypes (s-$CPDN_3$A-E) based on morphological similarity (Fig. 2E–I). These five $DN_3$ subtypes are comprised of 26 distinct cell types[9,10] and could thus be subdivided further in the future. During the revision of this manuscript, another study reported the existence of 12 $DN_3$ clusters/subtypes based on single-cell RNA sequencing[31], further supporting our analyses.

Additionally, we also identified several candidate $DN_{1p}$ which could not be classified into previously characterized $DN_{1p}$ cell types (Table 1). To verify if these neurons identified in the connectome are indeed $DN_{1p}$, we characterized them anatomically. Specifically, we performed multi-color flip-out (MCFO) analysis of $DN_{1p}$ using *Clk4.1M-Gal4* to elucidate the morphology of this densely packed neuronal cluster. In total, about 14 $DN_{1p}$ per hemisphere are labeled with the *Clk4.1M-Gal4* (Fig. 2J). MCFO analysis revealed $DN_{1p}$A (4-5 neurons) and $DN_{1p}$B (2 neurons) subtypes characterized previously (Table 1 and Fig. 2K). Our analysis additionally revealed the morphology of previously uncharacterized $DN_{1p}$ subtypes (Fig. 2L–N). $DN_{1p}$C and $DN_{1p}$E are each comprised of two neurons per hemisphere (Fig. 2L, N). $DN_{1p}$D is comprised of four neurons per hemisphere (Fig. 2M). Two of these project over the midline while the other two remain ipsilateral, suggesting that the $DN_{1p}$D is comprised of two morphologically distinct subtypes. Notably, our analysis confirmed that the candidate $DN_{1p}$ identified from the connectome are in fact morphologically similar to

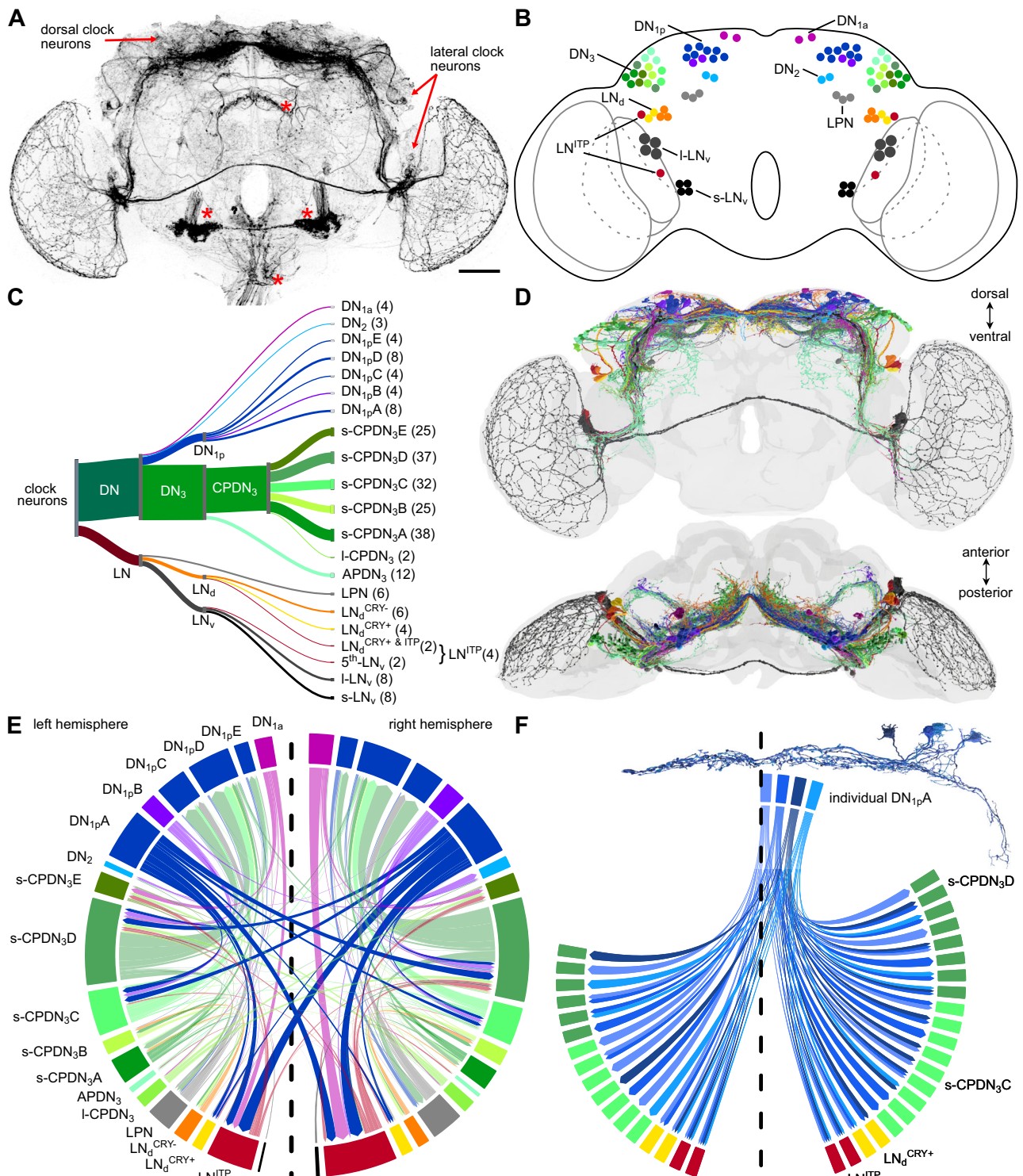

**Fig. 1 | *Drosophila melanogaster* circadian clock network. A** *Clk856-Gal4* drives GFP expression in most of the circadian clock neurons and some non-clock neurons (asterisk). Only a few DN$_3$ are included in this Gal4-line. Scale bar = 50μm. **B** *Drosophila* clock neurons can be divided into four classes each of Dorsal neurons (DN) and Lateral neurons (LN) based on their cell body location. These can be further subdivided into different cell types based on their morphology and gene expression. Subtypes of clock neurons are color-coded. This color code is used throughout the manuscript. **C** Classification and numbers of all clock neurons identified in the FlyWire connectome. Refer to Table 1 for details. **D** The morphology of identified clock neurons in the connectome largely resembles the

morphology of clock neurons marked by *Clk856-Gal4*. **E** Broad synaptic inter-connectivity between different cell types within the clock network. The direction of the arrow indicates the flow of information. Strong connectivity is observed from DN$_{1a}$ to LN$^{ITP}$, from s-CPDN$_3$C and s-CPDN$_3$D to DN$_{1p}$C-E, as well as from DN$_{1p}$A to LN$^{ITP}$, LN$_d$$^{CRY+}$, s-CPDN$_3$C and s-CPDN$_3$D. Note extensive contralateral connectivity between different cell types, most of which can be accounted for by the DN$_{1p}$A. The dashed line indicates the brain midline. **F** Individual DN$_{1p}$A in the right hemisphere forms both ipsilateral and contralateral connections with s-CPDN$_3$C, s-CPDN$_3$D, LN$_d$$^{CRY+}$, and LN$^{ITP}$. Source data for panels E and F are provided in the Source Data file. Brain mesh is from Dorkenwald et al., 2024.

**Table 1 | Identification and classification of *Drosophila* clock neurons in the FlyWire brain connectome**

| Clock Super Class | Clock Class | Clock Cell Type | Expected | Observed |
|---|---|---|---|---|
| Dorsal neurons (DN) | anterior $DN_1$ ($DN_{1a}$) | $DN_{1a}$ | 4 | 4 |
| | posterior $DN_1$ ($DN_{1p}$) | $DN_{1p}A$ | ~8-10 | 8 |
| | | $DN_{1p}B$ | ~4 | 4 |
| | | $DN_{1p}C$ | | 4 |
| | | $DN_{1p}D$ | ~12-14 | 8 |
| | | $DN_{1p}E$ | | 4 |
| | $DN_2$ | $DN_2$ | 4 | 3 |
| | $DN_3$ | small Central Projecting $DN_3$ (s-$CPDN_3$)A | | 38 |
| | | s-$CPDN_3$B | | 25 |
| | | s-$CPDN_3$C | | 32 |
| | | s-$CPDN_3$D | ~80 | 37 |
| | | s-$CPDN_3$E | | 25 |
| | | large Central Projecting $DN_3$ (l-$CPDN_3$) | | 2 |
| | | Anterior Projecting $DN_3$ ($APDN_3$) | | 12 |
| Lateral neurons (LN) | Lateral Posterior Neurons (LPN) | LPN | 6 | 6 |
| | dorsoLateral Neurons ($LN_d$) | Cryptochrome-negative $LN_d$ ($LN_d^{CRY-}$) | 6 | 6 |
| | | Cryptochrome-positive $LN_d$ ($LN_d^{CRY+}$) | 4 | 4 |
| | ITP-positive Lateral Neurons ($LN^{ITP}$) | Cryptochrome- and ITP-positive $LN_d$ ($LN_d^{CRY+ \& ITP}$) | 2 | 2 |
| | | 5th ventroLateral Neuron ($5^{th}$-$LN_v$) | 2 | 2 |
| | PDF-positive ventroLateral Neurons ($LN_v^{PDF}$) | large ventroLateral Neuron (l-$LN_v$) | 8 | 8 |
| | | small ventroLateral Neuron (s-$LN_v$) | 8 | 8 |
| **Total** | | | **~152** | **242** |

$DN_{1p}$C-E subtypes labeled by *Clk4.1M-Gal4*. Taken together, our identification and morphological characterization of novel $DN_3$ and $DN_{1p}$ subtypes provide a solid framework to comprehensively examine the connectivity of clock neurons.

The FlyWire connectome combines automatically detected chemical synapses with proofread neurons. These synapses represent an additional anatomical feature that could potentially distinguish neuronal groups. Consequently, we asked whether the classification of clock neurons based on differences in their synaptic connectivity aligns with the traditional anatomical and recent gene expression-based classification. To address this, we clustered clock neurons based on cosine similarity between their total synaptic inputs and outputs. Our clustering analysis shows that neurons of a given clock cell type (Table 1) usually cluster together, suggesting that neurons from the same group are more similar (in terms of synaptic connectivity) to each other than to other clock neurons (Supplementary Fig. 2A, B). For example, all three $LN_d^{CRY-}$ from one hemisphere are part of the same clade. Similarly, the two $DN_2$ are part of a clade. Exceptions to this are the clades containing $DN_{1p}$ and $DN_3$. For $DN_{1p}$, this can be explained by our findings which show that this group comprises five morphologically distinct subtypes. In the case of $DN_3$, while some of them form their own cluster, other $DN_3$ cluster together with different clock neuron subtypes. On one hand, this is not unexpected since it is unlikely for such a large group of neurons to have similar connectivity patterns. On the other hand, this is quite interesting since it provides insights into their possible function. For instance, heterogenous clusters comprising clock neurons of different classes have similar synaptic inputs and outputs, and may thus play similar roles in the clock network and beyond. Moreover, the connectivity-based clustering does not resolve the five subtypes of the s-$CPDN_3$. This suggests that these five s-$CPDN_3$ subtypes comprise synaptically heterogeneous cell populations. Nonetheless, synaptic connectivity-based classification of clock neurons largely aligns with the ones determined based on anatomical and gene expression differences.

Having identified all the clock neurons, we next sought to determine their synaptic interconnectivity which could facilitate intercellular coupling within the network. Generally, we regarded >4 common synapses per neuron as significant connections and >9 synapses as strong connections. We first wanted to validate our analysis by comparing it with previously reported connections. In agreement with previous reports[18,19], we observed strong synaptic connectivity from $DN_{1a}$ to $LN^{ITP}$, from $DN_{1p}A$ to $LN^{ITP}$ and $LN_d$, and from $DN_{1p}B$ to $DN_2$ clusters (Fig. 1E and Supplementary Fig.3), highlighting the robustness of our approach. Importantly, our analysis also uncovered novel connections between the different subgroups. Specifically, we observed strong contralateral and ipsilateral connectivity from $DN_{1p}A$ to $LN^{ITP}$, as well as additional significant connections with s-$CPDN_3$C, s-$CPDN_3$D, and $LN_d^{CRY+}$ across both hemispheres. Similarly, $LN^{ITP}$ also provide synaptic inputs to s-$CPDN_3$ clusters in both hemispheres (Fig. 1E and Supplementary Fig.3). This raised the question of whether $DN_{1p}A$ represents a heterogeneous population where one subgroup forms ipsilateral connections and the other contralateral. To address this, we examined their connectivity at cellular resolution (Supplementary Fig. 4, 5) which revealed that individual $DN_{1p}A$ indeed form both ipsilateral and contralateral connections (Fig. 1F). Our analysis thus identified $DN_{1p}A$ as an important center which links the clock network across the two brain hemispheres. These results are in line with previous reports of contralateral projections of $DN_{1p}[32]$. In contrast, there are virtually no synaptic connections between the s-$LN_v$ and the $LN_d$ neurons (Fig. 1E and Supplementary Fig.3-5), which control morning and evening activities, respectively. This is consistent with previous analyses using the hemibrain connectome[19].

To assess the extent of inter-individual differences in the numbers, neuronal projections, and synaptic connectivity of clock neurons, we next performed comparisons with the partial hemibrain connectome[33]. Several groups of clock neurons were previously identified in the hemibrain connectome including all s-$LN_v$, l-$LN_v$, $LN_d$, $LN^{ITP}$, LPN, $DN_{1a}$, and some $DN_2$, $DN_{1p}$, and $DN_3[17-19]$. Here, we identified additional $DN_{1p}$ and $DN_3$ (Supplementary Fig.6A–D). In total, 64 clock neurons can be identified in the hemibrain connectome, with the majority of missing neurons belonging to the $DN_3$ subgroups

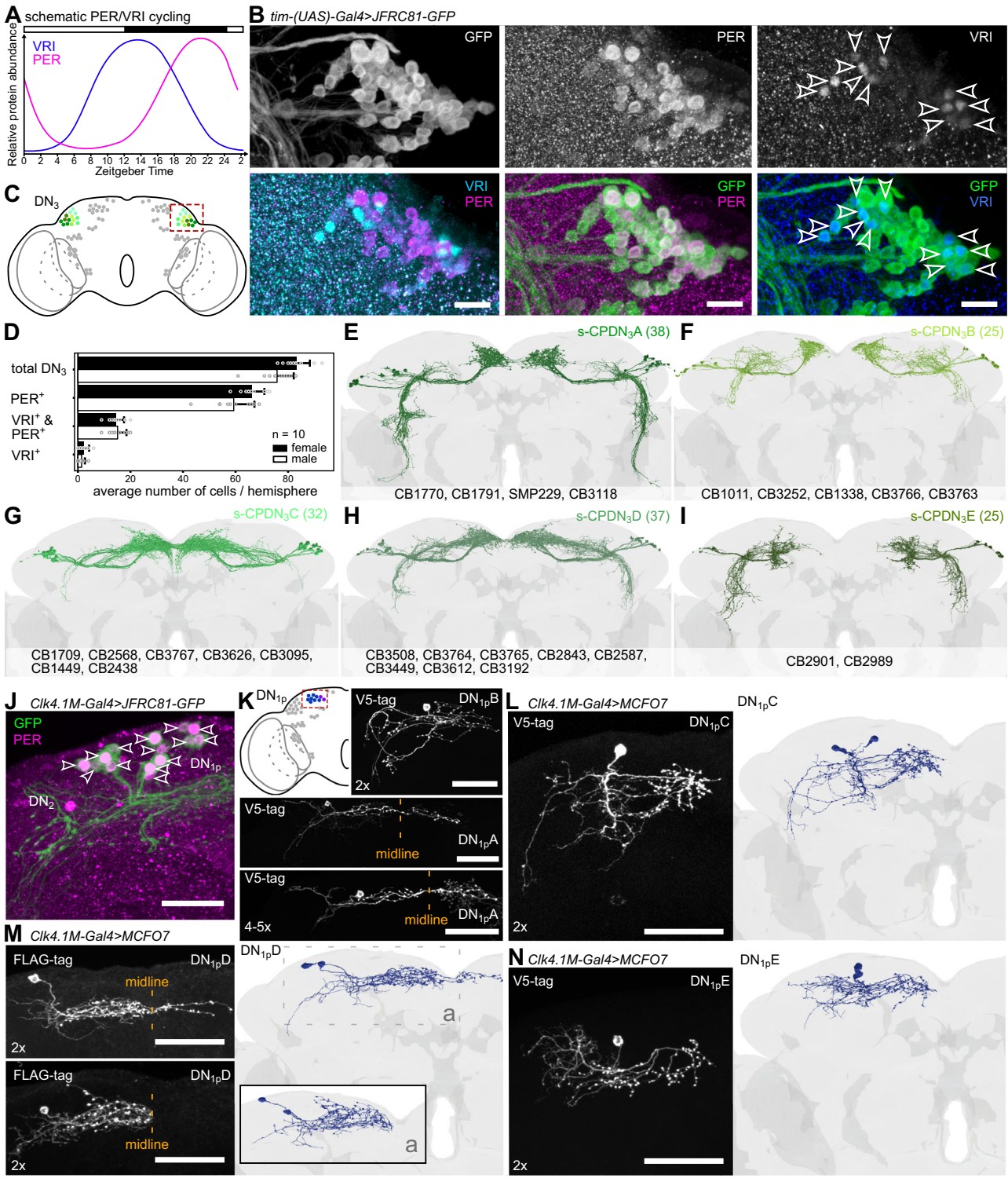

(Supplementary Fig.6A–D, Supplementary Data 2). Comparison of different subgroups revealed stable neuronal numbers across the two connectomes (Table 1 and Supplementary Fig.6D). Similarly, there is a high degree of stereotypy in the connectivity between the clock clusters (Supplementary Fig.6E, F). For instance, l-LN$_v$ and DN$_2$ form the least synaptic contacts with other clock clusters. At the opposite end of the spectrum, s-CPDN$_3$A are connected to all the clock clusters except for s-LN$_v$, l-LN$_v$, DN$_{1a}$, and DN$_{1p}$E. Given its partial nature, the hemibrain connectome lacks information about all contralateral connections, reiterating the significance of characterizing information flow across entire networks. Taken together, our analyses revealed hitherto

unknown connectivity between the clock neurons which could contribute to the robustness of the master clock. Moreover, the identification of the complete circadian neuronal network in the FlyWire connectome underscores the power of the fruit fly in pushing forward the frontier of our understanding of chronobiology.

**Validating clock neuron connectivity using trans-synaptic tracing**

While the FlyWire and hemibrain connectomes exhibit a high degree of stereotypy, we further used an independent approach to validate our connectivity analyses. We performed light microscopy-based trans-

**Fig. 2 | Morphology of newly identified s-CPDN$_3$ and DN$_{1p}$ subtypes. A** Schematic cycling of Period (PER) and Vrille (VRI) protein abundance in clock neurons[101]. **B** *tim-(UAS)-Gal4* drives GFP expression in all DN$_3$, many of which coexpress PER and about 12-19 coexpress VRI. When VRI staining is strong, PER staining is weak or not detectable. Image based on a female brain fixed at ZT1. Representative images are based on expression analyzed in 5 male and 5 female brains. Scale bar = 10 μm (**C**) DN$_3$ are closely associated with the lateral horn. **D** The number of DN$_3$ per hemisphere accounts for s-CPDN$_3$, l-CPDN$_3$, and APDN$_3$ identified in the connectome. On average, females have slightly more DN$_3$ (83 ± 5 standard deviation) compared to males (76 ± 6 standard deviation), which could be attributed to DN$_3$ being more densely packed (and thus difficult to quantify) in males. *n* = 10 hemispheres from 5 brains. Error bars depict standard deviation. **E**–**I** s-CPDN$_3$ cell types identified in the FlyWire dataset. Numbers in brackets represent the number of neurons in the total brain. s-CPDN$_3$ could be subdivided into five subtypes. Each subtype comprises two to eight different cell types that have unique morphological characteristics. **J** *Clk4.1M-Gal4* reliably drives GFP expression in 14-15 DN$_{1p}$ per hemisphere. **K** Multicolor flip-out (MCFO) analysis reveals previously characterized DN$_{1p}$A (4-5 neurons/hemisphere) and DN$_{1p}$B (2 neurons) subtypes. MCFO analysis additionally reveals the morphology of uncharacterized DN$_{1p}$ subtypes. DN$_{1p}$C and DN$_{1p}$E comprise two cells each per hemisphere **L**, **N**. DN$_{1p}$D comprises four cells per hemisphere two of which project over the midline while the other two do not, suggesting that the DN$_{1p}$D is comprised of two subtypes **M**. Scale bar = 50μm. Source data for panel **D** are provided in the Source Data file. Brain mesh is from Dorkenwald et al., 2024.

synaptic circuit tracing by expressing *trans*-Tango[34] using specific driver lines for different populations of clock neurons (Supplementary Figs. 7, 8). Upon driving *trans*-Tango with *Clk4.1M-Gal4* which labels most DN$_{1p}$ (Supplementary Fig. 7B), we observed post-synaptic signals in DN$_{1p}$, DN$_2$, DN$_3$, LPN, LN$_d$, 5th-LN$_v$, and s-LN$_v$ (Supplementary Fig. 9A). However, l-LN$_v$ and DN$_{1a}$ were not post-synaptic to DN$_{1p}$. Thus, our *trans*-Tango analysis of DN$_{1p}$ agrees with the connectivity of DN$_{1p}$ based on the connectomes (Supplementary Fig. 3 and Supplementary Fig. 6F). Similarly, a split-Gal4 line targeting DN$_3$ drives post-synaptic signals in DN$_{1a}$, DN$_{1p}$, DN$_2$, DN$_3$, LPN, LN$_d$, and l-LN$_v$ (Supplementary Fig. 9B), which mirrors the connectivity seen in the connectomes. While post-synaptic signal was not detected in most clock neurons of control flies, occasionally, a false post-synaptic signal was detected in two clock neurons (LN$_d$) from the entire network (Supplementary Fig. 10A). Hence, any potential synaptic output to LN$_d$ should be treated with caution. Overall, we observed similar congruency between the two approaches with other Gal4-lines including those targeting DN$_2$ (Supplementary Fig.10B), LPN (Supplementary Fig. 10C), and LN$^{ITP}$ (Supplementary Fig.10D).

Differences compared to the connectomes were observed when driving *trans*-Tango with *Pdf-Gal4* (for s-LN$_v$ and l-LN$_v$) (Supplementary Fig. 10E) and with the DN$_{1a}$-specific split-Gal4 line (Supplementary Fig.10F). In both cases, *trans*-Tango generated post-synaptic signals in more clock neurons than anticipated based on the connectomes. This discrepancy could be explained by: (1) the presence of additional neurons in the Gal4-line (e.g. PDF tritocerebrum neurons[35]), (2) daily remodeling of neural circuits, as shown previously for s-LN$_v$ and DN$_{1a}$[36,37] and/or (3) connections persisting through development. In summary, our *trans*-Tango analysis is largely in agreement with the clock network generated using the connectomes.

### Deciphering light input pathways to clock neurons

Following the successful validation of our connectivity data, we next identified all the major classes of neurons providing inputs to the clock network. To this end, we utilized the annotation scheme of our companion paper[9], which provides a hierarchical classification of all neurons in the connectome (Fig. 3A). We found that neurons intrinsic to the brain provide the majority of the inputs to the clock network (Fig. 3A–C). This includes visual centrifugal neurons projecting from the central brain to the optic lobes, visual projection neurons projecting from the optic lobes to the central brain, as well neurons intrinsic to the optic lobes and central brain (Fig. 3C). Examining inputs to specific clock clusters, we observed differential inputs across all the subgroups (Fig. 3B). As expected, s-LN$_v$ and l-LN$_v$ receive most of their input from optic lobe and visual centrifugal neurons as they have a large number of input synapses in the optic lobes and the accessory medulla (AME) (Supplementary Fig. 11). In contrast, APDN$_3$, l-CPDN$_3$, and LN$^{ITP}$ populations receive major inputs from visual projection neurons. The remaining clock clusters receive most of their inputs from central brain neurons (Fig. 3B, Supplementary Figs.11, and 12). In some cases, such as DN$_{1p}$C-E, DN$_2$, and s-CPDN$_3$A-E, a significant

portion of these central neurons are clock neurons themselves (Supplementary Fig.13), confirming prominent intercellular synaptic connectivity between some clock clusters. Interestingly, only 4 sensory neurons provide direct inputs to the clock network. These are anterior cells (aDT4) (Dr. Gregory Jefferis, personal communication) (Fig. 3A, D) which provide temperature inputs to LPN, DN$_{1p}$C, and DN$_{1p}$E[38,39]. Therefore, most of the strong thermosensory input to clock neurons is indirect via thermosensory projection neurons[40].

Having broadly classified the inputs from different neuronal super classes to the clock network, we probed further and identified individual cells providing the strongest synaptic inputs to clock neurons. For this purpose, we used a stringent threshold of 80 synapses to obtain a narrow list of candidate inputs. Our analysis discovered 13 neurons, including 7 aMe neurons (a subset of visual projection and visual centrifugal neurons) that are strongly connected to specific downstream clock neurons (Fig. 3E). For example, individual aMe3 and aMe6a neurons can form more than 79 synapses per neuron with APDN$_3$, while aMe8 are similarly connected to LN$_d$$^{CRY+ \& ITP}$ clock neurons (Fig. 3E). The unifying feature of these aMe neurons is their dense arborization in the AME and posterior lateral protocerebrum (Fig. 3E), where they anatomically interact with clock neuron dendrites (Supplementary Figs. 11 and 12)[18,41]. Interestingly, the aMe neurons themselves receive strong inputs from the extraretinal photoreceptors. Specifically, aMe3 and aMe6a neurons receive strong inputs directly from the Hofbauer-Buchner (HB) eyelets (Fig. 3E). Conversely, aMe8 receive indirect inputs from ocellar retinula cells via the ocellar ganglion neurons (OCG) type 2c (OCG02c) (Fig. 3E). Therefore, the clock receives strong light inputs from extrinsic photoreceptor cells, albeit indirectly. This is not surprising since light is the most important Zeitgeber for circadian clocks[42]. Flies synchronize their circadian clocks with the light-dark cycles using these extrinsic photoreceptor cells as well as via the blue-light photoreceptor Cryptochrome (CRY), which is expressed in about half of the clock neurons (Fig. 3F)[42]. While CRY interacts with the core clock protein Timeless and can quickly reset the clock, the different photoreceptor cells are important for sensing dawn, dusk, high light intensities, and day length, and for adapting morning and evening activities to the appropriate time of day. Regardless, we found little direct inputs from the photoreceptor cells and other sensory cells to the clock neurons (Fig. 3A). This is consistent with previous findings which revealed that most of the light input to the clock appears to be indirect[43]. In light of this and our *in-silico* circuit tracing analysis described above, we comprehensively characterized indirect connectivity between photoreceptor cells and clock neurons. Therefore, we traced all the disynaptic connections between them. Using the normal threshold of >4 synapses, we again recovered the strong connections from the H-B eyelets via the aMe3/aMe6a to the APDN$_3$, and additional weaker connections to the s-CPDN$_3$A, LN$^{ITP}$, l-LN$_v$, and LN$_d$$^{CRY+}$ (Fig. 3G). Furthermore, we revealed connections from R7/R8 compound eye photoreceptors to several clock neurons via aMe12[44] and other interneurons. While we did not observe any disynaptic connections from the ocellar retinula cells to

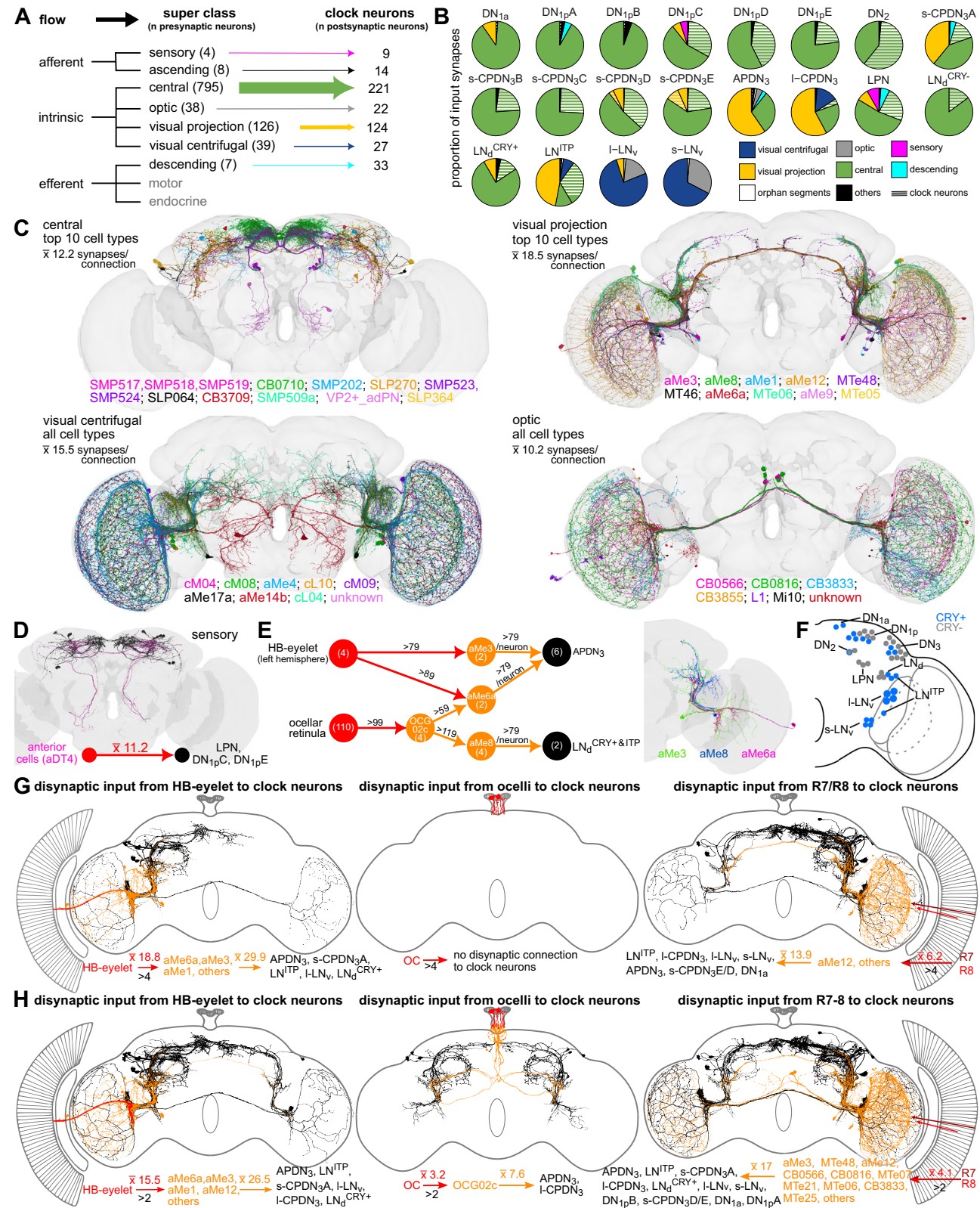

clock neurons using the normal threshold (Fig. 3G), reducing the threshold to >2 synapses revealed connections from the ocelli to APDN$_3$ and l-CPDN$_3$ via OCG02c (Fig. 3H). The synaptic connections from the ocelli to OCG and beyond are extensively characterized in our companion paper[10] and demonstrate interesting details that may also be valid for the other photoreceptor inputs to clock neurons. The majority of ocellar photoreceptors are synaptically connected to

ocellar ganglion neurons with thick axons (OCG01a-f) or directly to descending neurons (DNp28). These connections likely enable fast behavioral responses. In contrast, axons of OCG02c that connect to the clock neurons are rather thin and not suited for fast neuro-transmission. Instead, these neurons appear suited for collecting light information over time – a property needed for entraining the circadian clock. Further, collecting light information over larger time intervals

**Fig. 3 | Light and other inputs to the clock network. A** Input to the clock neurons grouped by the nine neuronal super classes annotated in the FlyWire connectome. Intrinsic neurons, specifically the central and visual projection neurons, are the largest groups providing inputs to the clock. **B** Breakdown of inputs to different types of clock neurons. **C** Neurons from the four different super classes (central, visual projection, visual centrifugal, and optic) which provide major inputs (based on the number of neurons) to the clock. **D** Anterior cells (AC, magenta) providing temperature inputs to LPN, $DN_{1p}C$, and $DN_{1p}E$. (black). **E** aMe neurons provide the strongest inputs to clock neurons. Upstream partners of aMe neurons with strong connectivity include HB-eyelet and OCG02c. Numbers above the arrows indicate the number of synapses and numbers in circles indicate the number of neurons. Note that the connection from the OC to OCG02c is below the threshold of 5 synapses/connection. Neuronal reconstructions of aMe3, aMe6a, and aMe8 are shown in one hemisphere. **F** Cryptochrome (CRY)-positive clock neurons are shown in blue. **G** Disynaptic inputs to the clock from the three types of extrinsic photoreceptors (HB-eyelet, ocelli, photoreceptor cells of the compound eye) using a threshold of >4 synapses. **H** Disynaptic inputs to the clock from the three types of extrinsic photoreceptors using a threshold of >2 synapses. For **C**, **D**, **G**, and **H**, numbers represent the average number of synapses. All cell types are listed according to their input strength (from high to low) to clock neurons. Note: HB eyelet was only identified in the left hemisphere. HB Hofbauer-Buchner, OC ocellar retinula cells, OCG ocellar ganglion cell. Source data for panels **B**–**D**, **G**, and **H** are provided in the Source Data file. Brain mesh is from Dorkenwald et al., 2024.

may not require a high synapse density. Thus, 3 to 4 synapses between retinula cells and the relevant downstream OCG observed here could be sufficient for this purpose (Fig. 3H). The same is also true for the photoreceptor cells of the compound eyes. Reducing the threshold of significant connections from 5 to 3 synapses revealed indirect clock input from additional photoreceptor cells, including those that project from the dorsal rim area of the eye (Fig. 3H). These photoreceptor cells are involved in polarized vision and might contribute to time-compensated sun compass orientation[45]. Whether the connectivity observed with a lower threshold of >2 synapses is functional in vivo remains to be seen; however, this is very likely since there are usually many photoreceptor cells that synapse onto only a few aMe neurons. For example, theoretically, the ~300 pale R8 cells project to only 3–4 aMe12 neurons[44], resulting in ~100 connections on average per aMe neuron. Even if each of these connections were mediated via only 3 synapses, each aMe neuron could potentially receive inputs from R8 cells via 300 synapses, which is quite substantial (Fig. 3H).

### Identification of clock network output pathways
Delineating the output pathways that translate daily 24-hour oscillations of the molecular clock into physiological and behavioral rhythms remains a major focus in chronobiology. Using the same strategy as above to identify the inputs, we systematically classified all the neurons downstream of the clock network. Most synaptic output from the clock network is directed to intrinsic brain neurons, and in particular, the central brain neurons (Fig. 4A–C). Except for l-LN$_v$, all clock clusters have a majority of their output onto central brain neurons. l-LN$_v$ mostly provide inputs to Medullary intrinsic neurons in the optic lobe (Mi 1, Fig. 4B, C), consistent with their role in adapting the sensitivity of the visual system to the time of day[46,47]. Further, examination of individual postsynaptic partners of clock neurons (Supplementary Fig. 14) reveals that the majority of the output from $DN_{1p}A$ is onto other clock neurons, namely LN$^{ITP}$, LN$_d$$^{CRY+}$, and s-CPDN$_3$C-D (Fig. 4B and D). s-CPDN$_3$C-D, in turn, provide strong output to DN1pC-E (Fig. 4D, E), thus linking CRY-negative $DN_{1p}$ clusters to CRY-positive $DN_{1p}$ clusters. After broadly classifying clock outputs, we next focused on specific cell types which receive the strongest synaptic inputs (>49 synapses) from clock neurons using an approach similar to the one used earlier for clock inputs. Our analysis identified the enigmatic Clamp neurons (Fig. 4F), which receive strong synaptic inputs from APDN$_3$. While the functions of most of these clamp neurons are still unknown, some of them output onto descending neurons, while others promote sleep[26]. Moreover, $DN_{1p}B$ provide strong inputs to Tubercle-innervating neurons (Fig. 4G), which are part of the anterior visual pathway[48]. Lastly, several clock neurons are strongly connected to diverse neurons from different neuropil regions (Fig. 4H).

The circadian clock is also known to modulate behaviors such as activity/sleep, spatial orientation, and learning and memory[49,50]. These behaviors are regulated by higher brain centers such as the central complex and mushroom bodies. The clock also influences systemic physiology via modulation of endocrine/neurosecretory cells

(NSC)[16,21]. Consistent with previous results[17,18], we found few direct connections from the clock to the central complex (Fig. 4I), mushroom bodies (Fig. 4J), and NSC (Fig. 4K). Consequently, we predicted that the clock output to these brain regions is either indirect or paracrine via neuropeptides. In line with this prediction, we found prominent disynaptic connectivity from clock neurons to central complex neurons (mainly fan-shaped body (FB) and ellipsoid body neurons (EB)), Kenyon cells (KC), dopaminergic neurons (DANs), mushroom body output neurons (MBONs) and NSC (Fig. 4I–K). Taken together, circadian modulation of neurons regulating diverse behaviors and physiology is largely indirect.

### Circadian modulation of behaviors via descending neurons
Next, we focused on clock output pathways to descending neurons which could influence diverse behaviors such as locomotion, feeding, and egg-laying. Interestingly, clock neurons provide direct inputs to 18 descending neurons (Fig. 5A). These descending neurons include those that have not yet been classified (Fig. 5B), as well as Allatostatin-C (AstC) and SIFamine (SIFa) peptidergic neurons (Fig. 5C), the latter of which modulate feeding, mating, and sleep[51]. When considering disynaptic connections, the connectivity between the clock network and descending neurons increased drastically, with approximately 24% of all descending neurons receiving indirect inputs from most of the clock neurons (Fig. 5A).

In order to obtain a better understanding of how the clock modulates descending neurons, and other downstream neurons in general, we first sought to characterize the neurotransmitters expressed in clock neurons. To address this, we used a combination of neurotransmitter predictions based on electron microscopy[52] and anatomical expression mapping of neurotransmitter markers (Fig. 5D–F), which are largely in agreement[53,54]. For some clock neurons such as $DN_{1p}A$ and LN$^{ITP}$, anatomical mapping (Fig. 5F) allowed us to fill the gap left by uncertain neurotransmitter predictions based on electron microscopy (Fig. 5D). Remarkably, most of the lateral clock neurons express the excitatory neurotransmitter acetylcholine, while most of the dorsal clock neurons express glutamate which can either be excitatory or inhibitory in *Drosophila*[52]. The only exception to these are the cholinergic l-CPDN$_3$. None of the clock neurons are predicted to be GABAergic. We also examined the neurotransmitters predicted to be expressed in presynaptic and postsynaptic partners of clock neurons (Fig. 5D). As seen for clock neurons, acetylcholine and glutamate appear to be the major neurotransmitters expressed in their synaptic partners. In addition to neurotransmitters identified here, previous work has revealed that s-LN$_v$ and l-LN$_v$ utilize the inhibitory transmitter glycine[55]. It is noteworthy that most of the postsynaptic partners of s-LN$_v$ and l-LN$_v$ are glutamatergic and cholinergic, respectively. Taken together, this comprehensive classification of neurotransmitters expressed in clock neurons allows valence assignment to the clock output pathways such as those to descending neurons.

We next utilized this information to characterize output pathways from the clock to downstream descending neurons. Direct synaptic output to descending neurons predominantly derives from

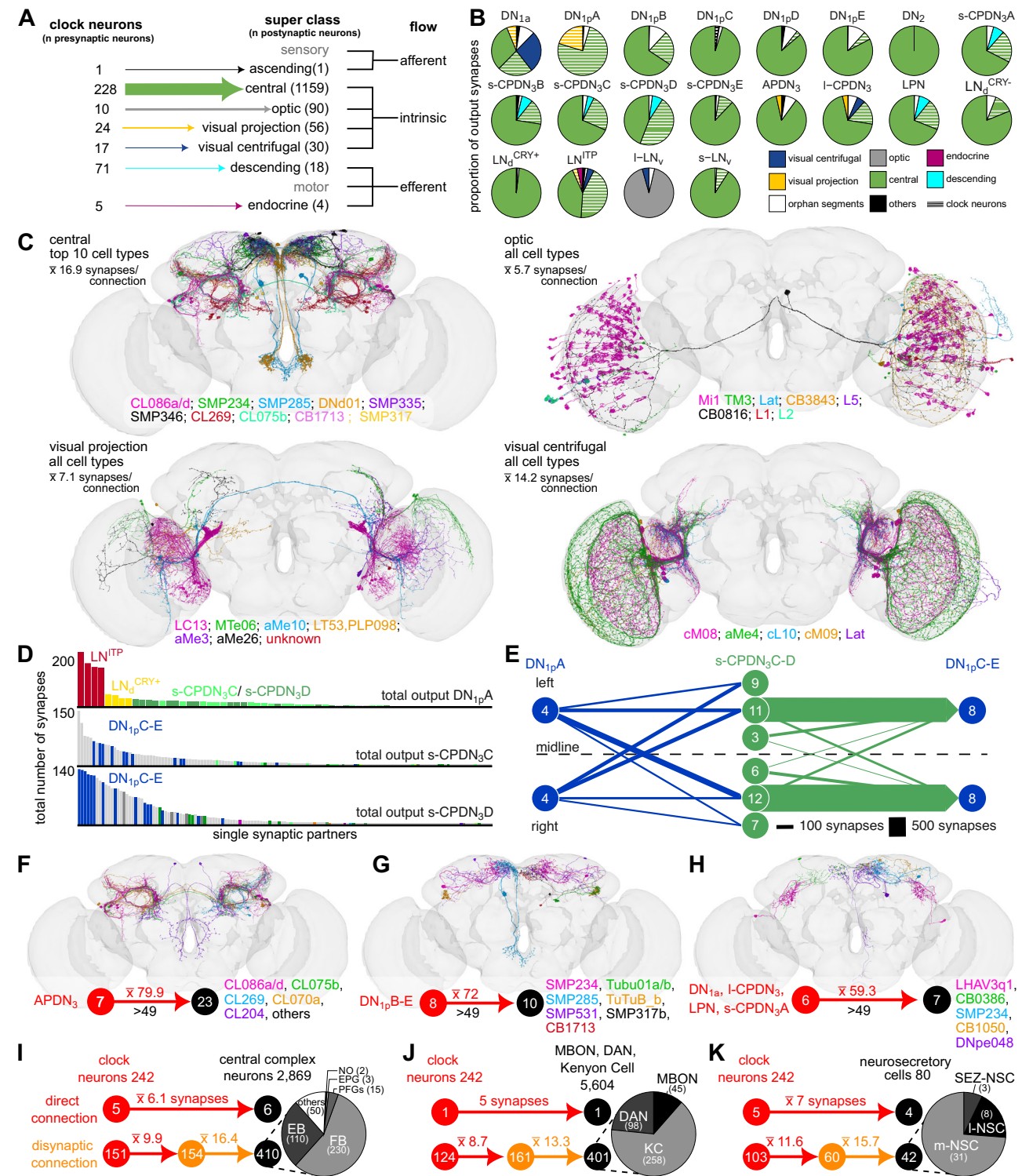

s-CPDN3A-D (glutamatergic), and LPN (cholinergic) (Fig. 5G). Three descending neuron types, namely DNpe048, DNpe033, and DNpe041, receive the majority of clock inputs. These descending neurons receive a substantial portion (about 25%) of their total input from clock neurons (Fig. 5H). We focussed further on DNpe048 which receives strong cholinergic and glutamatergic inputs from clock neurons. DNpe048 express crustacean cardioactive peptide (CCAP) and have recently been shown to regulate sugar and water ingestion by integrating inputs from other modulatory interneurons[56] (Fig. 5I). Allatostatin-A-expressing Janu-AstA neurons which regulate hunger and thirst[57] also provide inputs to DNpe048. It is thus tempting to speculate that

DNpe048 integrate time and temperature cues via s-CPDN3A-D and LPN along with hunger and thirst signals from diverse modulatory interneurons to regulate daily feeding rhythms (Fig. 5I, J).

Having elucidated major direct clock output to descending neurons, we decided to focus on the 312 descending neurons that receive clock inputs indirectly. These include oviposition descending neurons (oviDNa), vaginal plate opening descending neurons (vpoDN), and DNg30 that have previously been implicated in the regulation of reproductive behaviors. oviDNa regulate egg-laying[58], vpoDN control female sexual receptivity during courtship[59], and DNg30 express the neuropeptide natalisin which regualtes mating[60] (Fig. 5K). A recent

**Fig. 4 | Direct and indirect clock output pathways. A** Clock output grouped by the nine neuronal super classes annotated in the FlyWire connectome. Intrinsic neurons, specifically the central neurons, are the largest group of neurons downstream of clock neurons. **B** Breakdown of outputs from different types of clock neurons. All clock neurons, except for l-LN$_v$, provide a majority of their output to central neurons. l-LN$_v$ mostly provides inputs to optic neurons. **C** Neurons from the four different super classes (central, optic, visual projection, and visual centrifugal) which receive inputs from the clock. **D** Individual postsynaptic partners of DN$_{1p}$A and s-CPDN$_3$C-D are sorted based on the number of synapses. Postsynaptic clock neurons are colored based on their identity. **E** CRY-positive DN$_{1p}$A provide major inputs to CRY-negative DN$_{1p}$C-E via s-CPDN$_3$C-D. Contralateral connections from DN$_{1p}$A to s-CPDN$_3$ are stronger than ipsilateral connections. **F** APDN$_3$ forms the most output synapses in the clock network and is highly connected to several types of Clamp neurons (CL). **G** DN$_{1p}$B-E provides strong outputs to Tubercle-innervating

neurons. **H** Several clock neurons provide strong inputs to neurons from diverse neuropils. Mono- and di-synaptic clock inputs to neurons associated with the **I** central complex, **J** mushroom bodies, and **K** neurosecretory cells. Note that there are only a few direct connections from the clock to downstream higher brain centers. Numbers above the arrows indicate the number of synapses and numbers in circles indicate the number of neurons. Note: all numbers refer to neurons across both hemispheres. For **C**–**I**, cell types are listed according to the strength of the clock input (from high to low). EB ellipsoid body, FB fan-shaped body, NO noduli, EPG ellipsoid body-protocerebral bridge-gall neuron, PFGs protocerebral bridge glomerulus-fan-shaped body-ventral gall surround, KC Kenyon cell, DAN dopaminergic neuron, MBON mushroom body output neuron, SEZ-NSC suboesophageal neurosecretory cells, l-NSC lateral neurosecretory cells, m-NSC medial neurosecretory cells. Source data for panels **B**–**K** are provided in the Source Data file. Brain mesh is from Dorkenwald et al., 2024.

study elucidated the pathway from LN$_d^{CRY+}$ to oviDNa which controls rhythmic egg-laying[61]. Here, we show that glutamatergic DN$_{1a}$ and cholinergic l-CPDN$_3$ provide strong inputs to AVLP594 which in turn provide strong inputs to mating-regulating DNg30. Moreover, several types of clock neurons including s-CPDN$_3$C and APDN$_3$ provide strong inputs to vpoDN via pC1 neurons which integrate inputs regarding the female mating status[58,59]. Lastly, SIFa neurons, which receive both direct and indirect clock signals, also independently affect sexual receptivity. It remains to be seen if mating and sexual rhythms exist in *Drosophila*, and the contribution of these pathways towards them. In summary, our connectivity analysis indicates that the clock can have a major influence on diverse behaviors via outputs to descending neurons.

## Identifying the molecular basis of paracrine clock output pathways

Given the large repertoire of neuropeptides previously shown to be expressed in clock neurons[12,14,15], we predicted that peptidergic paracrine signaling is crucial in mediating intercellular coupling within the clock network as well as output to downstream neurons such as NSC. First, we determined if any additional neuropeptides are expressed in clock neurons. To address this, we used a publicly available single-cell transcriptome dataset of clock neurons[14] combined with immunohistochemical localization and T2A-Gal4 lines. Unsupervised clustering of all clock neuron transcriptomes using t-SNE analysis yields 32 independent clusters (data not shown), 16 of which have high expression of clock genes (*tim* and *Clk*) and can be reliably identified based on known markers (Fig. 6A, B). Our analysis revealed that most clock clusters express at least one neuropeptide (Fig. 6B). Consistent with previous studies, l-LN$_v$ express high levels of *Pdf*, whereas s-LN$_v$ express both *Pdf* and *short neuropeptide F* (*sNPF*) (Fig. 6B). Similar coexpression of neuropeptides is also observed in other clusters including the DN$_{1p}^{CNMa \& AstC}$ cluster which coexpresses *CNMamide* (*CNMa*), *AstC*, and *Diuretic hormone 31 (Dh31)* neuropeptides. In total, at least 12 neuropeptides are highly expressed in the clock network (Fig. 6B, C). Importantly, this includes novel clock-related neuropeptides, namely DH44 and Proctolin (Proc). *Dh44* is expressed in several clock clusters including DN$_{1a}$, DN$_{1p}^{AstA}$, DN$_{1p}^{sNPF}$, DN$_3^{VGlut}$, LPN and LN$_d^{NPF}$ (Fig. 6B, C). We independently confirmed the presence of DH44 peptide in these clusters using a combination of DH44 antibody or *DH44-T2A-Gal4* (Fig. 6D, Supplementary Fig. 15, and Supplementary Data 3). *Proc*, on the other hand, is strongly expressed in DN$_{1p}^{CNMa}$ and weakly in DN$_2$ clusters which was verified by driving GFP using a *Proc-T2A-LexA* driver (Fig. 6E and Supplementary Fig.15). *Proc* expression in other clock clusters such as LN$_d^{NPF}$ and DN$_3^{VGlut}$, remains to be validated. Lastly, we also detected *AstC* expression in additional clock neurons. AstC immunoreactivity was previously localized in DN$_{1p}$, DN$_3$, and LPN[62], which is in agreement with *AstC* transcript expression in DN$_{1p}^{Rh7}$ and DN$_{1p}^{CNMa \& AstC}$ clusters (Fig. 6B, C). Here, we show that AstC is additionally expressed in DN$_2$,

which were labeled using an antibody against VRI (Fig. 6B, F, Supplementary Fig.15, and Supplementary Data 3). Our expression analyses revealed the comprehensive neuropeptide complement of clock neurons (Fig. 6B–F, Supplementary Fig.15, and Supplementary Data 3) and provides the basis to explore paracrine targets of clock neurons.

As a first step in this direction and to validate our approach, we focused on select NSC which have been extensively characterized previously[23]. We predicted that NSC are targeted by clock-related neuropeptides since they receive sparse monosynaptic inputs from clock neurons despite being closely associated with them anatomically. To investigate potential paracrine signaling between clock neurons and NSC, we again turned our attention to single-cell transcriptomics. We identified single-cell RNA transcriptomes of medial NSC expressing DH44 (m-NSC$^{DH44}$) and insulin-like peptides (m-NSC$^{DILP}$), as well as lateral NSC expressing corazonin (l-NSC$^{CRZ}$), DH31 (l-NSC$^{DH31}$) and ITP (l-NSC$^{ITP}$)[63] based on previously identified markers, and quantified the expression of clock peptide receptors (Fig. 6G). Consistent with our prediction, our analysis indicated that multiple receptors for clock peptides are indeed expressed in different populations of NSC. Modulatory inputs to m-NSC$^{DILP}$ have been examined extensively and our analysis is in agreement with previous expression and functional studies[16,64–67]. Taken together, our analysis uncovers the molecular substrates of paracrine signaling between clock neurons and NSC.

We next explored the magnitude of peptidergic paracrine signaling between clock neurons themselves. To predict putative paracrine connections, we extensively mapped the expression of clock neuropeptide receptors within the clock network. Consistent with promoter and genomic fragment-based *Pdfr-Gal4* expression[68], *Pdf receptor* (*Pdfr*) transcript is highly enriched in most clock clusters (Fig. 7A, B), which we verified independently by expressing GFP using *Pdfr[RA]-T2A-Gal4* (Fig. 7C, D). Other receptors are more sparsely expressed within the clock network (Fig. 7A, B). However, most clock clusters express at least two receptors, with DN$_{1p}^{sNPF}$ expressing at least 7 receptors. We validated our single-cell transcriptome analysis by mapping the expression of select receptors using T2A-Gal4 knock-in lines. Our anatomical mapping of receptors (Fig. 7D and Supplementary Data 4) is largely in agreement with transcriptome data. In some cases, however, receptor mapping can provide additional insights. For instance, there are four transcript variants (A, B, C, and D) encoding the *Drosophila* Neuropeptide F receptor (NPFR). Using Gal4 lines specific for NPFR-A/C and NPFR-B/D isoforms, we showed that these isoforms are differentially expressed across the clock clusters, with A/C isoforms expressed more broadly than B/D isoforms (Fig. 7D and Supplementary Data 4).

Finally, we utilized our expression data of neuropeptides and their cognate receptors in clock neurons to delineate putative paracrine signaling pathways within the network. For this, we utilized an approach (Supplementary Fig. 16) similar to the one used recently to

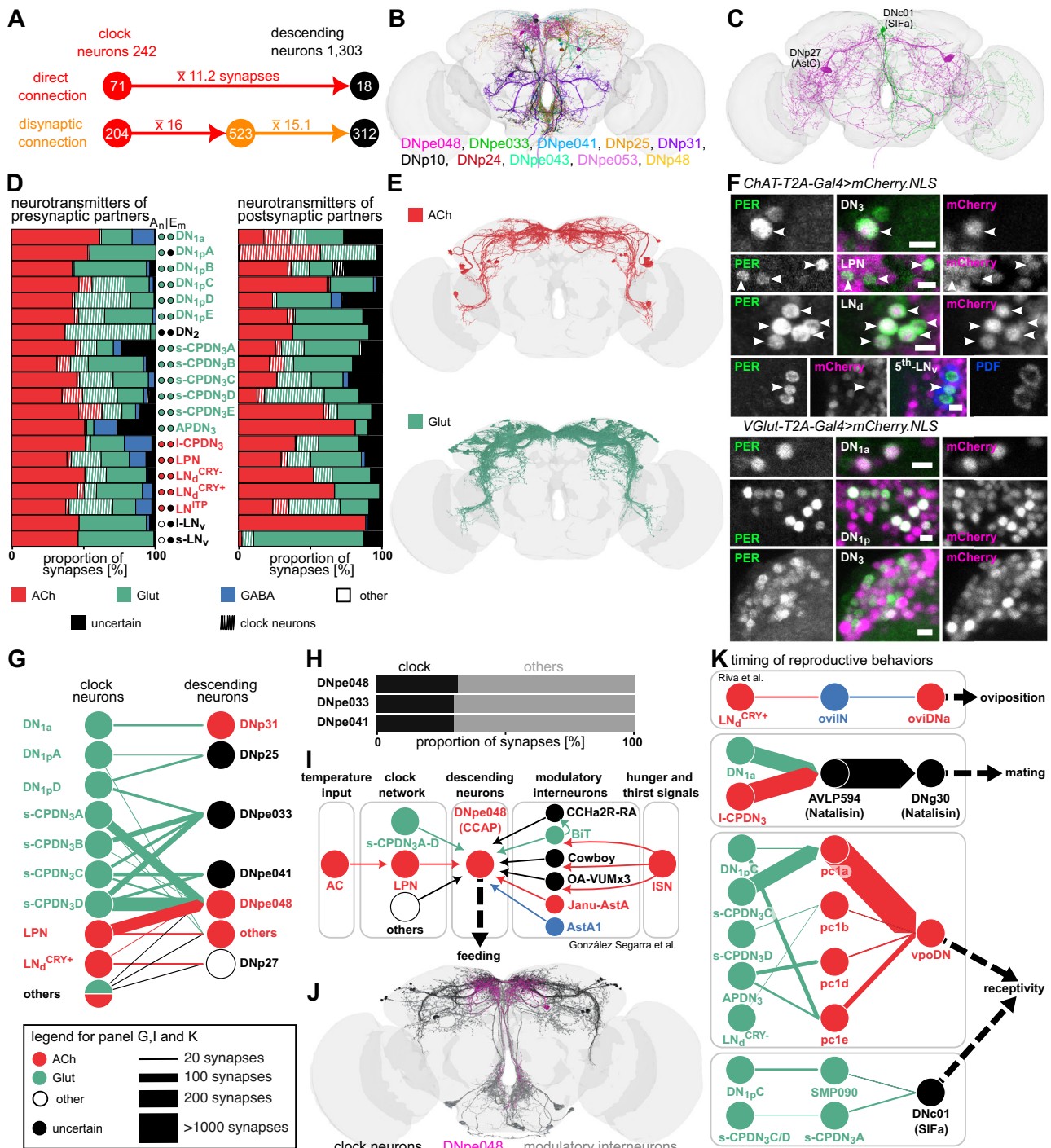

**Fig. 5 | Clock output to descending neurons. A** Mono- and di-synaptic clock output to descending neurons. **B** Novel and **C** peptidergic descending neurons receiving direct clock input. All cell types are listed according to the strength of the clock input (from high to low). **D–F** Neurotransmitters expressed in clock neurons and their pre- and post-synaptic partners. **D** Neurotransmitter prediction from Eckstein et al. 2024[52] based on electron microscopic data for presynaptic neurons of the clock network (left), the clock neurons (middle), and their postsynaptic neurons (right). **E** Most of the lateral clock neurons are cholinergic and most of the dorsal clock neurons are glutamatergic, while no clock neurons are GABAergic. Electron microscopy-based neurotransmitter predictions (Em) of clock neurons agree with **F** anatomical (An) expression mapping with T2A-Gal4 lines for neurotransmitter markers. **G** Weighted direct connections of clock neurons to descending neurons colored according to their neurotransmitter identity. s-CPDN₃

and LPN are strongly connected to three types of descending neurons. **H** Proportion of input synapses of the three descending neuron types that receive the strongest input from clock neurons. Each cell type gets about 25% input from clock neurons. **I** Schematic showing pathways via which CCAP-positive DNpe048 integrate time cues from the clock network with hunger and thirst signals to regulate feeding behavior. **J** Reconstructions of neurons that transmit time cues (black) and hunger and thirst-related signals (grey) to DNpe048 (magenta). **K** Pathways from clock neurons to descending neurons which could modulate the timing of reproductive behaviors. ACh acetylcholine, Glut glutamate, GABA γ-Aminobutyric acid, ChAT choline acetyltransferase, VGlut glutamate vesicle transporter, CCAP crustacean cardioactive peptide; Scale bar represents 5 μm. Source data for panels **A, D, E, G, H**, and **K** are provided in the Source Data file. Brain mesh is from Dorkenwald et al., 2024.

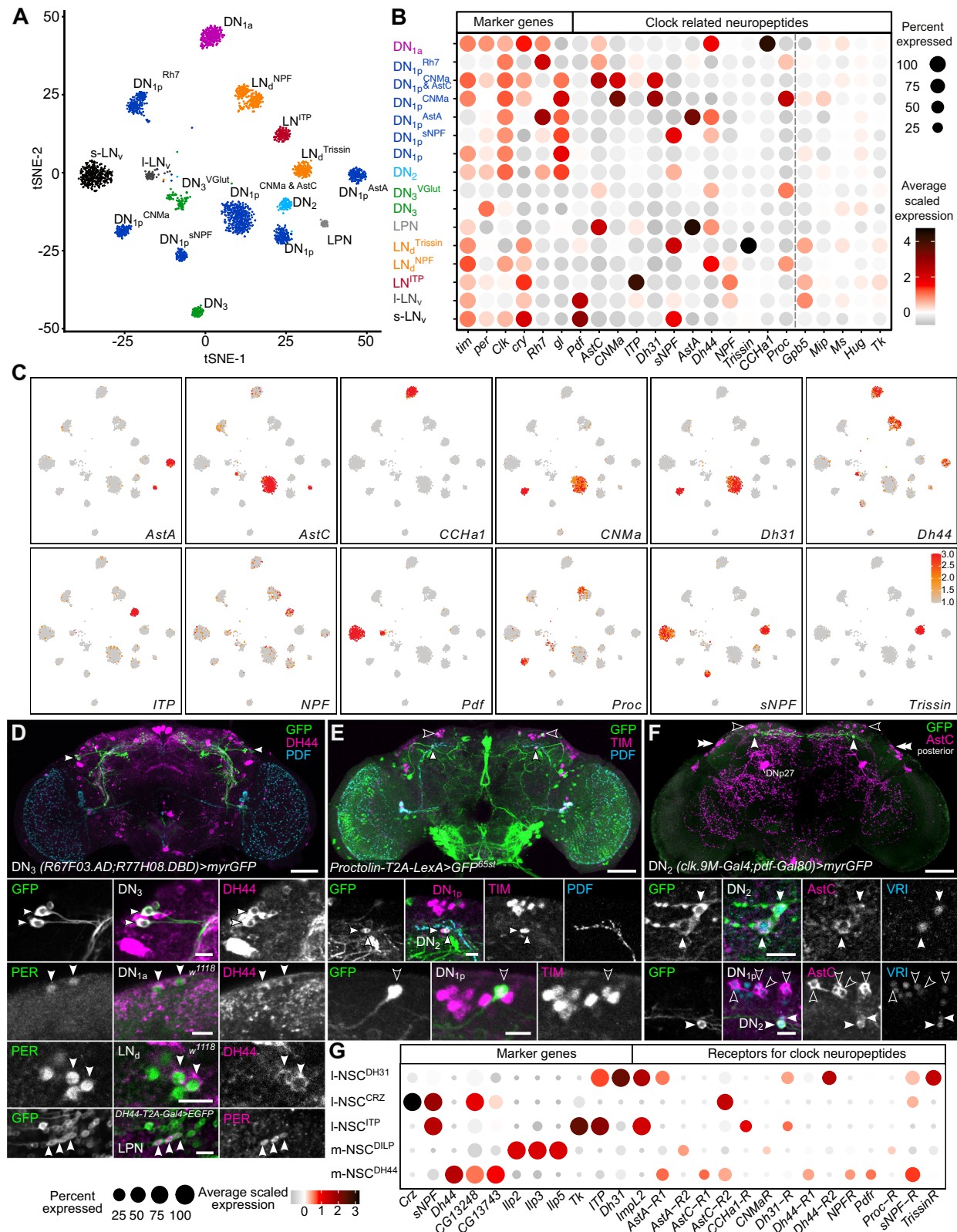

**Fig. 6 | Circadian clock neuropeptidome. A** Single-cell RNA sequencing of clock neurons reveals 16 distinct clock neuron clusters (shown in a t-SNE plot) that can be reliably identified based on known markers. **B, C** Clock neurons express at least 12 different neuropeptides. Based on average scaled expression (red = high and grey = low). **D** DH44 is a new clock-related neuropeptide which is expressed in APDN₃, DN₁ₐ, LN_d, and LPN (arrowheads). **E** Proctolin is a new clock-related neuropeptide which is expressed in one DN₁ₚ (open arrowhead) and two DN₂ (filled arrowheads). **F** AstC is expressed in DN₂ in addition to other clock neurons. Double arrowheads indicate DN₃ labeled by anti-AstC, filled arrowheads indicate DN₂ and open arrowheads indicate DN₁ₚ. Scale bars = 50 μm for overview and 10 μm for higher magnification images. PER Period, TIM Timeless, VRI Vrille, PDF Pigment dispersing factor, DH44 Diuretic hormone 44, AstC Allatostatin-C. **G** Single-cell transcriptomes of NSC express receptors for clock-related neuropeptides. Based on average scaled expression (red = high and grey = low). Source data for panels **A, C,** and **G** are provided in the Source Data file.

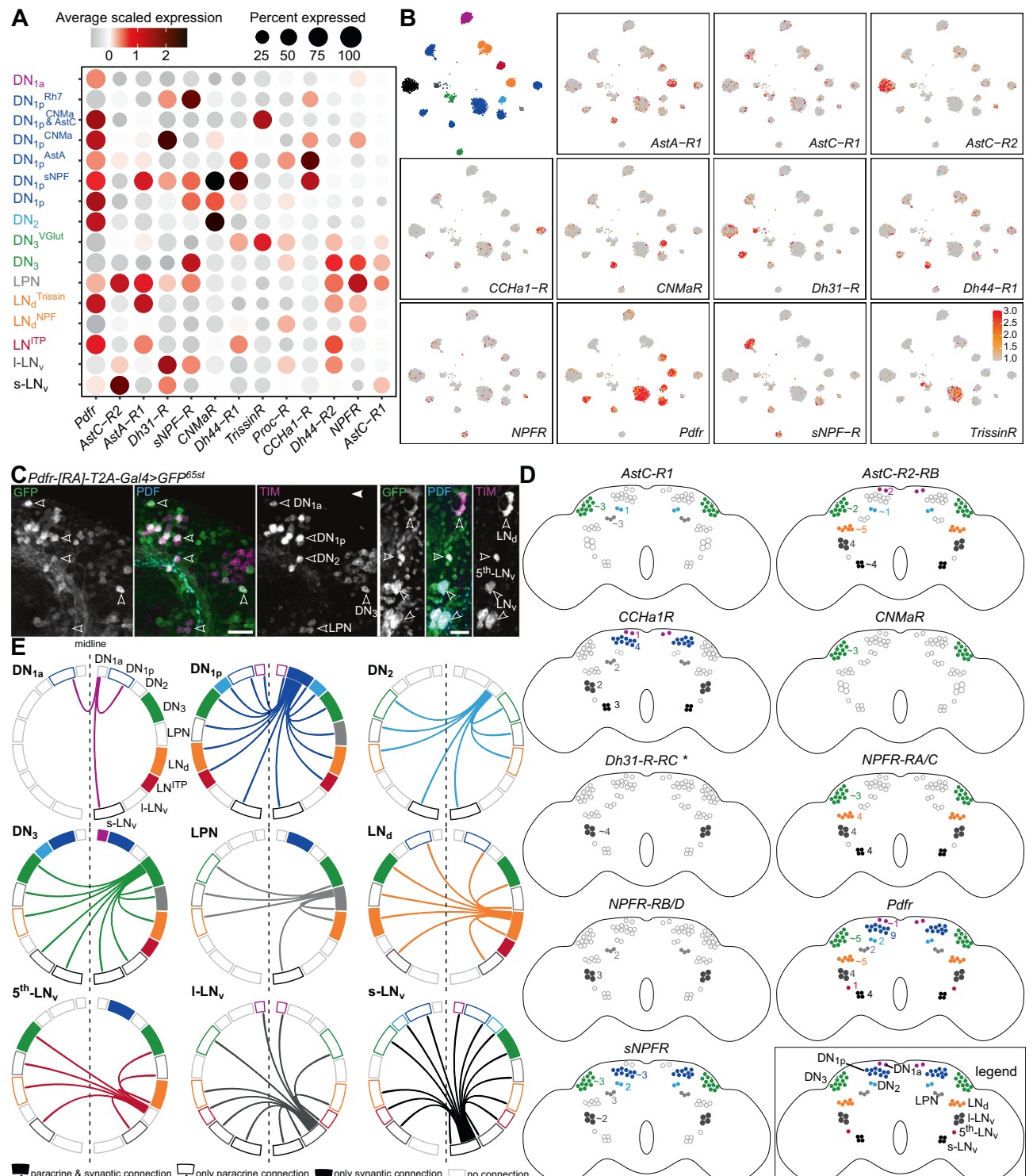

**Fig. 7 | Paracrine signaling within the clock network. A** Expression of clock-related neuropeptide receptors in different clock clusters was identified using single-cell transcriptome analysis. **B** t-SNE plots showing the expression of receptors in different clock clusters. Clustering is based on Fig. 6A. **C** *Pdfr[RA]-T2A-Gal4* drives GFP expression in all clock cell types. Arrowheads indicate different types of clock neurons that express PDFR[RA]. Scale bars = 20 μm. PDF, Pigment dispersing factor, TIM Timeless. **D** Schematics depicting expression of receptors in different clock neurons. The schematics are based on GFP expression using T2A-Gal4 lines for different receptors (see Supplementary Data 4 for confocal images). The

numbers refer to neurons expressing that receptor in one hemisphere. **E** Chord diagrams showing synaptic (filled boxes) and putative peptidergic paracrine connections (arrows) between clock neurons. This figure is based on synaptic connectivity in Fig. 1E, and peptide and receptor expression (following thresholding) mapping reported in Figs. 6B–F, 7A–D, Supplementary Fig. 15, and Supplementary Data 3, 4. See Supplementary Fig.16 for a detailed explanation of the filtering of putative paracrine connections. Source data for panels **A**, **B**, and **E** are provided in the Source Data file.

predict the *Caenorhabditis elegans* neuropeptide connectome[69]. Briefly, we used expression data based on independent methods to conservatively localize the expression of neuropeptides and their receptors across all the clock clusters. We also utilized the connectome to factor in the distance between neurons of different clusters. This was done to ensure that the cells releasing the peptide and those expressing its receptor are not further apart than a cut-off of 14μm, which was set based on a previous paracrine connectivity study[16]. Lastly, we only considered strong peptide-receptor interactions by disregarding ligands with $EC_{50}$ values for receptor activation higher than 500 nM. This stringent in silico approach allowed us to predict paracrine connectivity within the clock network with high confidence. Taking s-LN$_v$ as an example, this cluster expresses both PDF and sNPF. Following our expression thresholding, PDFR is expressed in most clock clusters, whereas sNPF receptor is expressed in DN$_{Ip}$, DN$_3$, LPN, and l-LN$_v$ (Fig. 7 and Supplementary Fig.16). Thus, in addition to providing synaptic inputs to DN$_3$, s-LN$_v$ can potentially provide paracrine inputs to most clock clusters across both hemispheres (Fig. 7E). Connectivity from l-LN$_v$ is also enhanced by paracrine signaling, although not to the same extent as it is for s-LN$_v$. Expanding this analysis to other clock clusters allowed us to comprehensively identify putative peptidergic signaling pathways between clock clusters (Fig. 7E). These pathways, however, can only be considered putative for three main reasons: (1) all clock neuropeptides are also expressed in other non-clock neurons[70], suggesting likely inputs from neurons extrinsic to the clock network, (2) we don't account for peptide efficacies and receptor affinities since these values were independently determined in distinct systems thus making comparisons difficult and (3) despite our best efforts to account for it, distance of peptide diffusion may vary. To put our peptide diffusion threshold of 14μm into perspective, we performed an additional stringent analysis where we lowered this threshold to 1μm. In this scenario, long-range peptide diffusion is not necessary and two neurons could communicate via paracrine signaling only if they are touching or almost within touching distance and express the peptide-receptor pair. Even using these highly stringent criteria, we see significant putative paracrine connections (~92% of those seen with the 14μm threshold) between different clock classes (Supplementary Fig. 16C). We speculate that the actual magnitude of paracrine signaling is somewhere between no peptide diffusion and brain-wide peptide diffusion, and our threshold of 14μm appears to be a good starting point to elucidate paracrine signaling pathways. In summary, peptidergic signaling greatly enriches the connectivity between different subsets of clock neurons. Additional investigations are necessary to determine which of these putative connections are functional in vivo.

## Discussion

### Peptidergic signaling supplements synaptic connectivity within the circadian clock network

Using a multipronged approach centered around the FlyWire connectome[9,10], we describe the whole-brain neural connectome of an animal circadian clock. Our analysis using the complete brain connectome eluded identification of one DN$_2$, and a couple of s-CPDN$_3$. Nevertheless, given the fact that we identified almost all of the expected clock neurons as well as several additional DN$_3$ (Table 1), our clock neuron synaptic wiring diagram is complete enough to be regarded as a connectome. Our clock connectome is also a significant upgrade (10-fold larger numerically) compared to the partial connectivity diagram based on the hemibrain connectome reported earlier[19]. The previous analysis was based on only 24 clock neurons and largely focused on LN clusters, while excluding l-LN$_v$, and several DN$_{Ip}$, DN$_2$, and DN$_3$ clusters due to the incomplete nature of the dataset. However, as evident from our analysis here, the DN in fact represents an important hub in the clock network and displays high synaptic connectivity. In particular, DN$_{Ip}$ plays a large role in clock cluster

interconnectivity and DN$_3$ appears important for clock output pathways. Our analysis also sheds light on the precise number of DN$_3$ in the clock network. Although approximately 80 DN$_3$ were previously estimated in the entire clock network, there are in total about 170 DN$_3$ based on our connectome and anatomical analyses. We anticipate that resources such as NeuronBridge[71], which allow for comparisons between electron and light microscopy datasets, will facilitate the identification of Gal4 drivers that target the novel s-CPDN$_3$ subtypes identified here. In addition, we also characterized the molecular basis for neuropeptide connectivity between the clock neurons, consequently highlighting putative peptidergic pathways within the clock network. Similar to vertebrates[22,72], the *Drosophila* clock network is highly peptidergic, with all clock neuron clusters expressing at least one neuropeptide. Notably, a majority of the clock clusters express two neuropeptides, and several express three, while the LPN expresses four neuropeptides. Similar neuropeptide coexpression is also evident in SCN neurons and thus appears to be a common feature of clock neurons[22]. Surprisingly, there is little to no overlap in the neuropeptide complement of the *Drosophila* and vertebrate clock neurons (Fig. 8). Orthologs of vertebrate clock neuropeptides including vasoactive intestinal peptide, arginine vasopressin, neuromedin S, cholecystokinin, gastrin-releasing peptide, and prokineticin 2 are either absent in the *Drosophila* genome or expressed outside the clock network. Hence, *Drosophila* and vertebrates have evolved to utilize different signaling molecules while still conserving the diversity of neuropeptide signaling within the clock networks. Remarkably, except for PDF, there appears to be little conservation in neuropeptide identities of clock neurons across different insects[49,73]. This suggests that it is more important to conserve the mode of communication (paracrine signaling) rather than the messenger (specific neuropeptide).

### Contralateral connectivity within the network prevents decoupling of clock neurons across the hemispheres

Analysis of interconnectivity within the clock network revealed extensive contralateral synaptic connectivity between the clock neurons, which is largely mediated by DN$_{Ip}$A and to a lesser extent by s-CPDN$_3$C, s-CPDN$_3$D, l-CPDN$_3$, and LN$^{ITP}$. Furthermore, paracrine peptidergic signaling amongst clock neurons has the potential to further strengthen this contralateral connectivity. Such a strong bilateral coupling of clock neurons could prevent the internal desynchronization of clock neuron oscillations between the two hemispheres – a phenomenon that can happen in other insects and even in mammals, but so far has not been observed in fruit flies[7]. In addition, consistent with previous analysis using the hemibrain connectome[19], we did not observe any synaptic connectivity between the s-LN$_v$ and the LN$_d$ neurons, which control morning and evening activities, respectively. In line with this observation, the phase relationship between morning and evening oscillators is plastic, consequently facilitating seasonal adaptations as in mammals[28]. Our data may also explain how the morning and evening oscillators in flies internally desynchronize under certain conditions[74]. One such relevant condition is increased PDF signaling during long days[21], which was shown to delay the evening oscillators[13,75,76] and may lead to internal desynchronization[77]. Here, we confirm the presence of PDFR in the evening neurons. Thus, enhanced PDF signaling could delay the evening neurons and bring them out of phase with the morning neurons, especially because the two sets of neurons are not connected via synapses. Taken together, the lack of interconnectivity between s-LN$_v$ and LN$_d$ could potentially be a factor promoting adaptation to different seasons and contexts.

### Light and other inputs to *Drosophila* and vertebrate circadian clocks

Our analyses reveal that extrinsic light input from the photoreceptor cells of the compound eyes, HB eyelets, and ocelli to the clock neurons

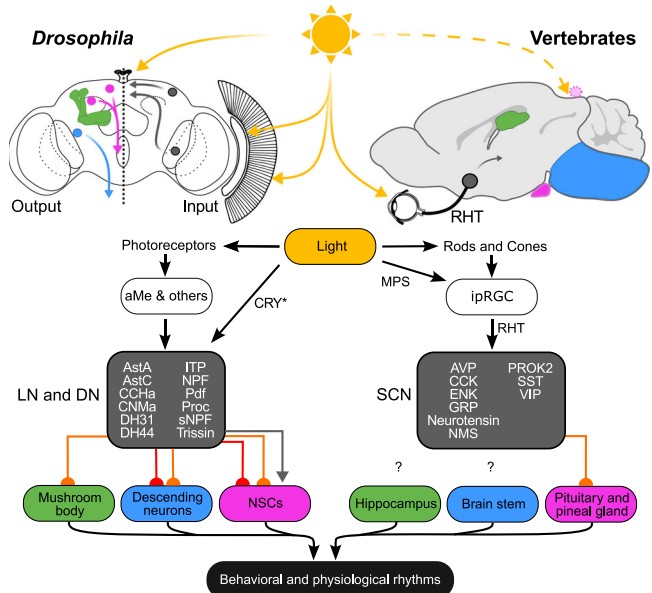

**Fig. 8 | Parallels in *Drosophila* and vertebrate clock input and output pathways.** Direct and indirect light input pathways to the *Drosophila* clock (comprised of LN and DN) and vertebrate suprachiasmatic nucleus (SCN). Note that the pineal gland receives light input only in non-mammalian vertebrates. *Drosophila* and vertebrate clocks utilize different neuropeptides (grey box). Output from the clock to downstream targets is either synaptic (direct in red and indirect in orange) or paracrine (grey arrow). CRY Cryptochrome, MPS Melanopsin, RHT Retinohypothalamic tract, ipRGC Intrinsically-photosensitive retinal ganglion cells, aMe Accessory medulla neurons, NSC neurosecretory cell, AVP arginine vasopressin, CCK cholecystokinin, ENK met-enkephalin, GRP gastrin-releasing peptide, NMS neuromedin S, PROK2 prokineticin 2, SST somatostatin, VIP vasoactive intestinal peptide.

is largely indirect, with the former two transmitting light inputs via aMe neurons. This situation may appear to be different from mammals where intrinsically photosensitive retinal ganglion cells in the retina project directly through the retinohypothalamic tract onto the SCN neurons (Fig. 8)[78]. However, even the mammalian clock receives indirect photoreceptor inputs from rods and cones via bipolar cells and retinal ganglion cells. Furthermore, indirect light inputs may reach the SCN also via the intergeniculate leaflets of the thalamus[79]. Altogether, this suggests that the fly and mammalian systems are not fundamentally different. One apparent difference is the presence of CRY, a cell-autonomous circadian photoreceptor, in subsets of *Drosophila* clock neurons that is sufficient for light entrainment in eyeless mutants[80]. Mammals lack light-sensitive CRY. Instead, they possess light-sensitive melanopsin in retinal ganglion cells, and mice lacking rods and cones can still entrain to light/dark cycles due to melanopsin[81]. Thus, flies and mice possess several redundant and partly parallel light-input pathways to entrain their clocks. The similarity is even higher when comparing flies with vertebrates in general. For instance, fish, birds, and reptiles possess additional photoreceptors in the pineal gland, which is reminiscent of the extraretinal HB eyelets or even the ocelli of flies (Fig. 8). Most importantly, vertebrates and flies use their eyes for both vision and entraining their circadian clocks, tasks that require completely different properties of light inputs. Vision requires image formation and fast neurotransmission, whereas circadian entrainment is dependent on integrating light collection over a longer time that can be at a slower rate. The connectome reveals that the number of synapses as well as the axon thickness of the neurons mediating this connectivity are very different between the two types of photoreception. Hence, they are aptly suited to perform their required functions.

## Limitations of our approach

The synaptic and putative paracrine connections reported here can only be considered predictions until they are functionally verified. In the case of paracrine connectivity, functional connectivity experiments (in normal and peptide mutant flies) using electrophysiological methods or genetically encoded secondary messenger sensors (calcium and cAMP) are needed to confidently establish functional paracrine connectivity. Additionally, the synaptic connectivity reported here is likely an underestimation due to several factors: (1) we did not explore connectivity via gap junctions, (2) proofreading of photoreceptor axons was not complete in the version utilized here[10], and (3) we generally used a connectivity threshold of >4 synapses. Preliminary expression analysis using the single-cell transcriptomes of clock neurons suggests that gap junction genes are enriched in the clock network (data not shown) and they can influence activity-rest rhythms[82]. It remains to be seen which clock neurons are additionally coupled via gap junctions and how this electrical connectivity complements synaptic and peptidergic connectivity detailed here. Moreover, as discussed earlier, fewer than 5 synapses could also represent functional connections which were largely disregarded in our analyses. Further, during the revision of this manuscript, additional synapses were predicted in the FlyWire connectome by Princeton University. Some of these additional synapses increase the connections between clock neurons, specifically those mediated by LN[ITP] (not shown). Crucially, these additional synapses and connections do not change the conclusions reported here. While the FlyWire and hemibrain connectomes exhibit a high degree of stereotypy, they are both based on an adult female brain. The lack of a male brain connectome currently prevents any comparisons on sex-specific differences within the circadian network and its output pathways which could influence sexually dimorphic behaviors and physiology. Further, the connectome provides a singular snapshot of connectivity which could change depending on the time of day, the age of the animal as well as its internal state.

## Conclusion

In conclusion, our circadian clock connectome, the first of its magnitude, is a significant milestone in chronobiology. Given the high conservation of circadian network motifs between *Drosophila* and vertebrates, this connectome provides the framework to systematically investigate circadian dysregulation which is linked to various health issues in humans including sleep, metabolic, and mood disorders. Moreover, it will also facilitate the development and experimental validation of novel hypotheses on clock function.

## Methods

### Fly strains

*Drosophila melanogaster* strains used in this study are listed in Supplementary Table 1. T2A-Gal4 and T2A-LexA knock-in lines generated previously [83,84] were obtained from the Bloomington *Drosophila* Stock Center (BDSC) and Dr. Shu Kondo. Flies were maintained under LD12:12. Flies for peptide and receptor mapping were reared at 25 °C on *Drosophila* medium containing 0.7% agar, 8.0% glucose, 3.3% yeast, 4.0% cornmeal, 2.5% wheat embryo, and 0.25% propionic acid. Flies for *trans*-Tango and Gal4 verification were raised at 18 °C and 25 °C, respectively, on a standard medium containing 8.0% malt extract, 8.0% corn flour, 2.2% sugar beet molasses, 1.8% yeast, 1.0% soy flour, 0.8% agar and 0.3% hydroxybenzoic acid. 2-week-old flies were used for *trans*-Tango analysis.

### Immunohistochemistry and confocal imaging

**Neuropeptide and receptor mapping.** Immunostainings were performed as described previously[85]. Briefly, male flies were entrained in LD12:12 at 25 °C for at least 3 days. Whole flies sampled at Zeitgeber time (ZT) 20 were fixed in 4% paraformaldehyde in phosphate-

buffered saline (PBS) with 0.1% Triton X-100 (PBS-T) for 2.5 h at room temperature (RT). Fixed flies were washed three times with PBS, before dissecting their brains. Samples were washed with PBS-T three times. The samples were blocked in PBS-T containing 5% normal donkey serum for 1 hour at RT and subsequently incubated in primary antibodies at 4 °C for 48 h. Following six washes with PBS-T, the brains were incubated in secondary antibodies at RT for 3 h. Lastly, the samples were washed six times in PBS-T and mounted in Vectashield mounting medium (Vector Laboratories, Burlingame, CA, USA). At least five brains for each strain were used for immunostaining to characterize the clock cells stained by anti-GFP antibodies (Gal4 or LexA expression) in the first experiment and the clock cells were briefly characterized with PDP1 and PDF antibodies. In the second experiment, we conducted the same immunostaining with anti-TIM antibodies, but only for positive strains, to confirm the prior results. Images were taken from at least three different brains using laser scanning confocal microscopes (Olympus FV1200 and FV3000, Olympus, Tokyo, Japan), and analyzed using Fiji.

**Mapping DH44 and AstC expression in clock neurons, verification of clock neuron Gal4 lines, and *trans*-Tango analysis.** Flies for PER staining were fixed at ZT23 (1 hour before lights-on) since PER levels are maximal at this ZT[86]. Flies for VRI staining were fixed at ZT 20. Immunohistochemistry was performed as described previously[17,18]. Samples were scanned with a Leica TCS SP8 confocal microscope equipped with a photon multiplier tube and hybrid detector. A white light laser (Leica Microsystems, Wetzlar, Germany) was used for excitation. We used a 20-fold glycerol immersion objective (0.73 NA, HC PL APO, Leica Microsystems, Wetzlar Germany) for wholemount scans and obtained confocal stacks with 2048 × 1024 pixels with a maximal voxel size of 0.3 × 0.3 × 2 μm and an optical section thickness of 3.12 μm. For noise reduction, we used a frame average of 3. Images were acquired using Leica Application Suite X (v3.5.7.23225) and analyzed using Fiji. All the primary and secondary antibodies are listed in Supplementary Table 2.

**Multi-color flip-out.** Multi-color flip-out (MCFO) analysis was performed to identify the morphology of the 8 so far uncharacterized $DN_{1p}$[87]. *Clk4.1M-Gal4* flies were crossed to MCFO7 flies. The F1 generation was kept at 25 °C and 1-7 days old male and female flies were used for staining. Immunohistochemistry was performed as described above. All the primary and secondary antibodies are listed in Supplementary Table 2.

**$DN_3$ quantification.** To quantify the number of $DN_3$ in the adult brain, *tim-(UAS)-Gal4 > JFRC81-GFP* male and female flies were entrained at 25 °C for 3 days and then fixed at ZT1. Brains were stained for GFP, PER, and VRI as described above. Images were acquired using a Leica TCS SP8 confocal microscope as described above. A 63-fold glycerol immersion objective (1.3 NA, HC PL APO, Leica Microsystems, Wetzlar Germany) was used to acquire detailed images of the $DN_3$ cluster. Images were recorded at 1024 × 1024 pixels with a maximum voxel size of 0.18 × 0.18 × 0.33 μm and an optical section thickness of 0.99 μm. Cells were counted using the Multi-Point tool in Fiji. 5 brains each for males and females were used for quantification.

**Connectome datasets and neuron identification.** For all analyses, we used the v783 snapshot of the FlyWire whole-brain connectome and its annotations (annotations last updated: 26.12.2023)[9,10]. We also used the adult hemibrain connectome (v1.2.1) for comparisons (Scheffer et al. 2020). The pipeline for identifying clock neurons is summarized in Supplementary Fig.1. Briefly, neurons were first identified based on morphological and/or connectivity features described previously[17,18,26,30,32], or based on NBLAST similarity to identified neurons in the hemibrain[9]. Morphology clustering (v630) for 124,988

FlyWire neurons[9] was then used to identify neurons that were morphologically similar to the clock neurons identified above. Several clock neuron types (e.g. $APDN_3$, s-$LN_v$, $LN_d$, etc.) formed their own morphology cluster, suggesting that additional neurons similar to them do not exist in the dataset. If a morphology cluster comprised more neurons besides the clock neurons identified earlier, all the additional neurons were considered putative clock neurons (Supplementary Fig.1), which were subsequently filtered based on different features. First, the cell body position was determined by mapping the coordinates of the nuclei[88] to the root IDs of the neurons. The average distance between the cell body position of neurons in one clock cluster was determined per hemisphere and candidate neurons laying less than twice the average distance away were retained for manual comparison of morphological features and connectivity described in previous studies. In addition, hemilineage and cell type information was used to determine whether the candidate neurons could be additional clock neurons. If new clock neurons were identified, the whole procedure was repeated. By definition, a cell type is a uniquely identifiable neuron in the dataset[9]. Based on our analysis, if a certain cell type was part of a clock cluster, all other neurons of that cell type were also part of this clock cluster. Thus, the likelihood of other cells that have a similar morphology to the clock neurons identified here is extremely low. FlyWire and Hemibrain cell IDs of identified clock neurons are provided in Supplementary Data 1 and 2, respectively.

**Paracrine connectivity prediction.** For paracrine connectivity analysis, peptide and receptor expression was first determined based on single-cell RNA sequencing analyses[14]. A gene was considered to be expressed in a given cluster if 1) it was expressed in more than 49% of the cells within that cluster and 2) the expression level was above the set threshold. A threshold of 0.208 average scaled expression was used for peptides and 0.067 for receptors to align single-cell RNA sequencing data with previously established expression data. Further, T2A-Gal4 expression patterns for peptides and receptors were analyzed in conjunction with antibody staining. This information, together with published antibody staining and live imaging data for receptors (see Source Data for Supplementary Fig. 16), was used to generate expression matrices based on the different methods (Supplementary Fig. 16A). To reduce the number of false positives, we only considered the expression of peptides when shown by two independent methods. A receptor was deemed present if its expression was demonstrated using two independent methods or based on live imaging data (Supplementary Fig. 16A). Next, we calculated the distance between neurons of different clock clusters based on their skeletons (v783_L2). To calculate a threshold for the distance that a peptide can diffuse, we used the closest distance (14 μm) between the s-$LN_v$ and the m-$NSC^{DILP}$ since paracrine PDF signaling between these neurons has been demonstrated previously[16]. If two clock clusters included neurons that were located closer than this distance, we assumed that peptidergic paracrine signaling was possible between these clusters (Supplementary Fig. 16B). Additionally, some *Drosophila* receptors are activated by multiple ligands. Hence, based on the previously reported $EC_{50}$ values for receptor activation (see Source Data for Supplementary Fig. 16), we set a stringent threshold of 500 nM to determine if a peptide can activate a receptor. Lastly, to identify putative paracrine connections, we combined the distance matrix with the peptide-receptor expression matrix (Supplementary Fig. 16C). Additionally, we used a distance threshold of 1 μm for a more stringent analysis.

**Neurotransmitter predictions.** Neurotransmitter predictions are based on Eckstein et al. (2024)[52]. As suggested by Eckstein et al. (2024), we only considered neurotransmitters that were predicted with more than 62% certainty. Further, we followed Dale's principle[89] and Lacin's law[90]. Hence, we assume that acetylcholine, glutamate, and GABA are not coexpressed, and all neurons belonging to the same hemilineage

express the same neurotransmitter. We only considered the fast-acting neurotransmitters (i.e., acetylcholine, glutamate, and GABA) for our analyses as their predictions were the most reliable. If a neurotransmitter was not predicted for a given neuron, we used predictions from neurons of the same cell type and hemilineage for which only one type of neurotransmitter was predicted. If several neurotransmitters were predicted for the same cell type/hemilineage, we did not use this information to classify neurons with uncertain predictions. Lastly, we used T2A-Gal4 lines for *ChAT*, *VGlut*, and *VGAT* to examine the expression of these neurotransmitter markers in clock neurons. We used neurotransmitter identities determined based on these anatomical data in cases where neurotransmitter predictions were not possible based on the electron microscope data.

**Data visualization.** Data was visualized using ggplot2 (v 3.4.2, Wickham, 2016) and circlize (v 0.4.15) for R (v 4.2.2) in RStudio (2022.12.0)[91]. Reconstructions were downloaded using the navis library (v 1.0.4, https://github.com/navis-org) and cloud-volume library (v 8.10.0, https://github.com/seung-lab/cloud-volume) for python (v 3.8.5), and visualized using blender (v 3.01, Community, B. O. 2018). For visualizing a large number of neurons, Codex (doi: 10.13140/RG.2.2.35928.67844) and FlyWire neuroglancer were used[92].

**Connectivity analyses.** Connectivity data was analyzed using the natverse libraries (v 0.2.4) for R in RStudio[93]. Synaptic connectivity similarity for clock neurons was analyzed based on all their input and output synapses. Cosine similarity analyses were conducted with coconat (v 0.1.1.9; https://github.com/natverse/coconat) for R. Filtered synapses were retrieved in Python using the navis and pandas (v 1.1.3[94],) libraries. Unless stated otherwise, a connection with more than 4 synapses was considered significant.

**Single-cell transcriptome analyses.** Expression of neuropeptides and their cognate receptors in clock neurons was mined using the single-cell RNA sequencing dataset and analysis pipeline established earlier[14]. NSC transcriptomes were identified from the brain transcriptomes generated previously[63]. The parameters used to identify the different cell types were based on previous studies and provided below[64,65,70,95–99].

DH31 (6 cells): ITP > 2 & Dh31 > 4 & amon > 0 & Phm > 0.
DH44 (12 cells): Dh44 > 1 & CG13248 > 1 & CG13743 > 0 & Lkr > 0
CRZ (4 cells): Crz > 3 & sNPF > 3 & Dh44 = 0 & ITP = 0 & ChAT = 0 & Gr64a = 0
ITP (7 cells): Tk > 1 & sNPF > 1 & ITP > 1 & ImpL2 > 1 & Crz = 0
IPC (19 cells): Ilp2 > 3 & Ilp3 > 3 & Ilp5 > 3

Expression was scaled based on all genes in the dataset. All analyses were performed in R-Studio (v2022.02.0) using the Seurat package (v4.1.1[100]).

**Statistics and Reproducibility.** All micrographs shown in the figures are representative images based on at least five independent samples.

### Reporting summary
Further information on research design is available in the Nature Portfolio Reporting Summary linked to this article.

## Data availability
Connectivity analyses can be performed using the cell IDs provided at https://codex.flywire.ai/ and with the scripts provided (see below). Source data for plots are provided with this paper. Source data are provided with this paper.

## Code availability
Code used for the analyses and data visualization is publicly available on GitHub: https://github.com/Zandawala-lab/Synaptic-connectome-of-the-Drosophila-circadian-clock_Reinhard-and-Fukuda-et-al.-2024.

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

## Acknowledgements

We would like to thank Selina Hilpert and Barbara Mühlbauer for technical assistance, Maria Steigmeier for preliminary investigations on PDFR expression, and Tatsuya Yokosako for generating the LN$^{ITP}$ split-Gal4 line. Stocks obtained from the Bloomington *Drosophila* Stock Center (NIH P40OD018537) were used in this study. We are also grateful to Drs. Fumika Hamada for *Clk.9M-Gal4;pdf-Gal80*, Shu Kondo for T2A-Gal4 lines, Paul E. Hardin for VRI antibody, Ralf Stanewsky for PER antibody, Jadwiga Giebultowicz for TIM antibody, Justin Blau for PDP1 antibody, Heinrich Dircksen for ITP antibody, Susan Morton for RFP antibody, and Jan Veenstra for AstC and DH44 antibodies. We thank the Princeton FlyWire team and members of the Murthy and Seung labs, as well as members of the Allen Institute for Brain Science, for the development and maintenance of FlyWire (supported by BRAIN Initiative grants MH117815 and NS126935 to Murthy and Seung). We also acknowledge members of the Princeton FlyWire team and the FlyWire consortium, especially Drs. Gregory Jefferis, Sven Dorkenwald, and Philipp Schlegel for troubleshooting and guidance. Special thanks to Austin T Burke and Mareike Selcho from the FlyWire consortium for contributing >10% effort towards editing and proofreading at least 10% of clock neurons. We are also thankful to Drs. Theresa McKim and Dick Nässel for helpful feedback during the preparation of this manuscript, as well as the Division of Instrumental Analysis, Okayama University for the laser scanning confocal microscopes (FV1200 and FV3000). M.Z. was supported by funding from the University of Würzburg, Deutsche Forschungsgemeinschaft (DFG; ZA1296/1-1), and NV INBRE grant from the National Institute of General Medical Sciences (GM103440). D.R. (DFG; RI 2411/1-1) and C.H.F. (DFG; FO 207/16-1) were supported by

DFG. T.Y. was supported by JSPS (KAKENHI 19H03265). A.S. and M.S. were supported by the OU fellowship (JST SPRING, Grant Number JPMJSP2126). We also acknowledge funding from the DFG for the Leica TCS SP8 microscope (251610680, INST 93/809-1 FUGG).

## Author contributions

N.R., C.H.F., T.Y., and M.Z. conceived the study. N.R., D.R., C.H.F., T.Y., and M.Z. supervised the project. N.R., A.F., G.Manoli, E.D., A.S., G.Möller, M.S., and M.Z. performed the experimental work and analyzed the data. N.R. and M.Z. performed computational analyses. M.Z. wrote the manuscript with input from C.H.F and N.R. All authors read, provided feedback, and approved the final manuscript.

## Funding

## Competing interests

The authors declare no competing interests.
