## [Transparent Peer Review file · Nature Communications]

Synaptic connectome of the *Drosophila* circadian clock

Corresponding Author: Dr Meet Zandawala

Version 0:

Reviewer comments:

Reviewer #1

(Remarks to the Author)

The author's response to two rounds of critiques is to respectfully disagree with my concerns. I think the authors have two options. 1.) If the work is to appear in a journal like Nature Communications, they must follow-up the new anatomical features they have characterized with experimental work to support the conclusions they make and remove the highly speculative sections of the study, particularly the speculations regarding non-synaptic peptidergic connectivity, which amount to reckless speculation presented as results. 2.) They could simply describe the interesting anatomical insights that their new connectomic analysis has provided, avoid the over interpretation of these results, discuss the possible implications of the new connections they have discovered, and publish the paper in a journal more appropriate for a work of neuroanatomy. Option one would require years of work. Option 2 would be a welcome and well cited addition to the literature that would be the basis for a great deal of future work. There are excellent journals that would be appropriate for such a study.

Below, I respond to the each of major points made by the authors, which are placed in quotations below.

"Accurate quantification of the total number of DN3 clock neurons for the first time which resulted in the identification of 90 additional clock neurons than previously estimated (Figure 2A-I). These additional novel neurons provide the major output to descending neurons that regulate behaviors, thus providing novel insights into how rhythmic behaviors are generated (Figure 4B and new figure 5)."

I concede that the authors revision of the total number of clock neurons is important and significant. But this observation is immediately followed by pure speculation that "these additional novel neurons provide the major output to descending neurons that regulate behaviors, thus providing novel insights into how rhythmic behaviors are generated." Absent experiments that support this conclusion, this is simply speculation presented as confirmed results.

"We identified for the first time the morphology and connections of all DN1p (including the Cryptochrome-negative ones) and classified them into 5 subgroups (the CRY-positive DN1pA and B, and the CRY-negative DN1pC-E). So far, only the function of some CRY-positive DN1p is known. The function of the other DN1p is unknown but according to our analyses, they are a major contributor to the clock output, and are predicted to play diverse roles based on their morphology and connectivity."

While it is of interest to see the connectomic analysis extended to new subclasses of clock neuron, the authors once again resort to pure speculation to make the anatomical details take on a significance that is not yet established, going so far as to conclude that the DN1ps "...are a major contributor to the clock output." Given that DN1ps are dispensable for most aspects of circadian timekeeping and sleep regulation, it's not clear how the authors can conclude this based on the new connections they have characterized. This has been my concern from the beginning with this study. Interesting new anatomical details are being used to make conclusions that would require a great deal of work to confirm.

"While it was known that DN1pA project contralaterally, their downstream synaptic partners were only partially known. Our analysis convincingly shows that these contralateral projections are nearly exclusively to other clock neurons which suggests a mechanism for clock synchronization across the two hemispheres (Figure 1E-F). This is in contrast to other clock

neurons (e.g., LNd and DN3) which also project contralaterally but do not provide extensive inputs to other clock neurons. Hence, morphological and connectivity analyses do not always correlate.”

I agree that DN1 connectivity was only “partially known.” This was true for all neuronal classes and is still true to some extent: as comprehensive as this study attempts to be, new details will surely emerge as the clock network continues to be examined. Furthermore, I agree that the differential patterns of connectivity displayed by the various classes of clock neuron are of interest. But this analysis simply extends the work previously done, including a substantial amount of work done by the authors themselves. The idea that the pattern of contralateral connections formed by the DN1pAs “suggests a mechanism for clock synchronization across the two hemispheres” is complete speculation. Given that glass loss of function mutants, which lack DN1ps entirely appear to have no problem creating coordinated rhythms would appear to be strong evidence against this hypothesis, as they provided no behavioral evidence that the two hemispheres are not synchronized.

“One of the major partners of the DN1pA are the here newly described small central projecting DN3 (s-CPDN3 C-D). The s-CPDN3 C-D project on the CRY-negative DN1pC-E, which are main output neurons of the clock network and synapse on different central brain neurons. The function of these central brain neurons is still unknown but several of them make connections in the subesophageal ganglion and the superior medial protocerebrum and are suited to modulate these brain regions in a circadian manner.”

Again, this is interesting, and it provides some new insight into the synaptic connectivity underlying the clock neuron network. But the authors are, once again, taking a bit of new information and then speculating about the possible functional implications of these connections. This is fine of course, but I fear that the study repeatedly attempts, in describing the results, to make claims that it can't make without additional mechanistic work. Anatomical studies are valuable to the field and can be presented on their own, with along with some speculation about the functional import of the new anatomical features revealed being presented more cautiously in the discussion. It is important here to note that clock neurons have been shown to have non-circadian functions (e.g., PMID 22561453), so output pathways from clock neurons may underly non-circadian functions.

“We have also performed comprehensive trans-Tango-based circuit tracing analyses within the clock network to complement the connectome analyses (Figure S9-10).”

This is interesting, but, in addition to the synapses detected in the adult connectome, trans-Tango will reflect synapses that form during development and subsequently disappear along with synapses that are present during the first days of adulthood that no longer exist in the mature adult. So, in my opinion, the authors are likely obscuring the clear picture provided by the new connectomic analysis with the use of this technique.

“We have also discovered additional neuropeptides expressed in the clock neurons using two different approaches (single-cell RNA sequencing analysis and immunohistochemistry) (Figure 6). The functions of these neuropeptides within the clock network will be characterized in a future study.”

Again, an excellent start to what is likely to be an insightful line of questioning. An observation warranting description in a high quality journal that features neuroanatomical studies. Interesting to the field, but we must await mechanistic experiments to reveals what, if anything, these peptides do.

“Additional analysis to show that s-CPDN3 neurons function as integrators by linking different subclasses of DN1ps (new Figure 4E). This is important because the newly described CRY-negative DN1pC-E have outputs to still unknown central brain neurons that may be involved in the transfer of clock signals to other brain regions and beyond.”

This is interesting, and I agree that the DN3 work is the most novel aspect of the work.

“Fast-acting neurotransmitter expressed in all the clock neurons based on EM-based neurotransmitter prediction as well as immunohistochemical analyses. We show that glutamate and acetylcholine, but not GABA are the major neurotransmitters in clock neurons (new Figure 5D-F). The inclusion of valence to the synaptic connectivity will facilitate future studies modeling clock-driven behaviors.”

This is also speculation. The EM predictions and GAL4 expression supporting these conclusions amount to an intriguing set of observations that I hope will be followed up with functional studies. Absent follow-up studies, these observations remain speculative and are better suited for a journal that features anatomical work.

“A more detailed analysis of how the circadian system is connected to descending neurons. Specifically, we show that s-CPDN3 and LPN provide major direct outputs to descending neurons (new Figure 5G).”

Here the authors offer more details on output pathways from clock neurons. This is of interest and may end up being the basis of future experimental work, but does not represent major new findings that move the field forward.

“We postulate pathways via which the clock regulates rhythmic feeding and reproductive behaviors to highlight the utility of this resource in generating novel hypotheses that can be experimentally tested (new Figure 5H-K).”

Such postulation is a welcome part of the discussion of the potential functions of the output pathways characterized, but this does not amount to a “novel insight,” as stated by the authors.

“Lastly, we agree with this reviewer that the connectome of Pars Intercerebralis would be of great interest and deserves to be pursued in a greater depth. Since reviewer 2 also suggested to reorganize the manuscript and make it less overwhelming, we have now removed our identification of neuroendocrine cells from the manuscript. This allows us to focus more on clock neurons (modified Figure 4 and new Figure 5).”

I am happy to see that the authors agree with something that I've expressed.

“It is not clear to the authors which previous connectomic studies have shown light and other sensory inputs to clock neurons. The hemibrain data did not include the optic lobe and the ocelli. Similarly, putative inputs from other sensory modalities (e.g., temperature) were also not examined in previous connectomic studies. Hence, our study is the first to highlight all the pathways from sensory neurons to clock neurons.”

Work from the Jefferis lab has described thermal inputs onto the DN1as and DN1ps (PMID: 32619476), work in the Fernandez lab has characterized inputs onto the lateral neuron classes (PMID: 35766361), including inputs from the HB eyelet and accessory medulla. The authors themselves has examined inputs, including temperature inputs onto the DN1ps (PMID: 35574472) and have described synaptic inputs onto the LPNs (PMID: 34961936).

“The only similarity between our study and the two companion connectome papers in the FlyWire package is that they looked at ocelli pathways. Our analyses complement their analyses and go beyond as we also focus on light input from structures other than the ocelli. The companion paper only examined the connectivity from the ocelli to ocellar ganglion cells. This is in contrast to our analysis where we show how the light input via the ocelli could travel to the clock neurons. Hence, we don't see any redundancy in our analysis and the other connectome studies. Nonetheless, since we are using the connectome and broad annotations described in the two companion connectome papers, we feel that it is appropriate to cite them where relevant.”

I agree that the inclusion of the ocelli is a most welcome addition to the literature and a source of novelty in the present study.

“We agree with the reviewer that we, as a scientific community, do not know much about how far peptides can diffuse through the brain. We have also acknowledged this issue in the manuscript. Nonetheless, our analyses used to determine putative peptidergic connections are a step above previous studies utilizing just expression data (see: Figure 2 Supplement 4 in <https://elifesciences.org/articles/65745>). We showed putative peptidergic connectivity under 2 situations....We think that the actual amount of paracrine signaling is somewhere in between the two extremes and our threshold of 14 micro diffusion distance seems a good starting point to examine this mode of signaling.”

I remain deeply skeptical of this aspect of the study. We simply do not understand non-synaptic peptidergic signaling in the intact brain well enough to present anything like the “putative peptidergic connectivity” presented in this paper. This is a central question of great importance that will require a great deal of work examining the physiology of such connections and the anatomical rules that govern peptide diffusion through the brain. This aspect of the work amounts to reckless speculation in my opinion and should not be published without functional studies, even with the “third [more conservative] situation” added by the authors.

In conclusion, I think there is much of importance and interest in the new anatomical features described in this study, but there is simply too much speculation and over-interpretation, and this remains a serious weakness of the study.

Reviewer #2

(Remarks to the Author)

The revised manuscript by Reinhard, Fukuda, and colleagues has significantly improved compared to their original submission to Nature, after multiple rounds of revisions. This study has made several major findings that will be of great interest to the fly circadian field:

1. This study expanded the number of clock neurons from approximately 150 to 242 by adding and characterizing new members of the mysterious DN3 and DN1 subgroups.
2. The research mapped out the connectivity within the circadian neural network and detailed the input/output profiles of circadian neurons.
3. Notably, the additional analysis presented in this version (Figure 4 DE and Figure 5) provides several clear roadmaps for future research on the output pathways of circadian regulation of various behaviors, such as feeding, drinking, and mating.
4. By combining the EM prediction dataset (Eckstein et al., 2024) and the scRNAseq dataset (Ma et al., 2021), the study has assigned neurochemical labels (molecular basis) to the connectivity they mapped.

Together, these findings assemble a comprehensive circadian clock connectome. This will become an important resource for the fly circadian field. I did not find any other major concerns and look forward to exploring the dataset.

Version 1:

Reviewer comments:

Reviewer #3

(Remarks to the Author)

As requested, I'm commenting mostly on the authors' response to Reviewer 1. Overall, the authors have addressed these comments (see exceptions below). And I believe they are justified in not doing additional experiments to support their predictions. Some amount of speculation is acceptable although this should be of a general nature and refrain from specific claims, which can only come from experimental evidence.

I note though that the author response to the reviewer's comment about trans-Tango is not correct. The reviewer noted that trans-Tango can be misleading as it may report synapses that occurred earlier in life. In response, the authors said that they used 2 week old adults for trans-Tango. However, as long as the relevant transgenes were expressed all through development (which appears to be the case), they may still be detecting synapses that occurred at earlier stages of life. This should be acknowledged. It may even explain why some of their trans-tango experiments detected more synapses than indicated by the connectome.

Also, I was confused by the following response regarding the authors' efforts to trace sensory projections to clock cells: "We apologize for the sloppiness in our response in the rebuttal letter. In fact, the Jefferis lab (Marin et al., 2020) has conducted a connectome analysis of the whole adult female brain (FAFB) to uncover the connectivity of the antennal glomeruli involved in temperature and humidity sensing. This study identified indirect thermosensory input pathways to clock neurons, which was the reason we focused mainly on direct thermosensory sensory input to clock neurons. We have now added a short sentence (in italics) to the manuscript clarifying this: "This suggests that the clock receives strong light inputs from extrinsic photoreceptor cells, albeit indirectly. S i m i l a r l y , t h e r m o s e n s o r y i n p u t t o c l o c k n e u r o n s i s a l s o i n d i r e c t v i a t h e r m o s e n s o r y p r o j e c t i o n n e u r o n s (M a r i n e t a l . , 2 0 2 0) ."

Further, we have cited a personal communication from Gregory Jefferis confirming that the only four sensory neurons that provide direct input to clock neurons are thermosensory neurons. Nevertheless, the other three studies, including our own (Shafer et al., 2022; Reinhard et al., 2022a,b) have used the hemibrain that lacks the optic lobes and ocelli. Therefore, the present study is actually the first to show all pathways from the photoreceptor cells to the clock neurons. The surprising and novel finding of the present study is that most of the photoreceptor input to the clock neurons is indirect. This does not exclude the direct input from the HB eye cells to the clock neurons described by us and others. However, this direct input is very weak compared to the massive indirect input via accessory medulla neurons. Therefore, we consider it extremely important to describe these indirect connections in detail."

It seems that in the first para they are saying thermosensory input is indirect, and in the second that Greg Jefferis has shown direct input. Also, if this was already done, then why does the first para say "we focused mainly on direct thermosensory sensory input to clock neurons"? The manuscript also did not clarify this issue.

Summary of major findings

Circadian clocks are universal. This fundamental biological system orchestrates various physiological and behavioral processes in response to daily environmental changes. Understanding the neural networks and signaling pathways that drive these processes is crucial to uncover the mechanisms behind daily rhythms and their impact on overall organismal health. We utilized a multipronged approach centered around the FlyWire connectome to generate the first neural connectome of an animal circadian clock. We demonstrate novel insights into clock functioning and regulation, and our research has uncovered several noteworthy findings. Specifically, we:

- **discovered all neurons within the *Drosophila* circadian clock within the FlyWire connectome**, including those that have been difficult to identify and characterize using other anatomical approaches.
- **accurately quantified the total number of DN₃ clock neurons** and showed that they are significantly more abundant than previously estimated.
- **reveal that the *Drosophila* clock network is comprised of at least 240 neurons** as opposed to the previous rough estimate of 150 neurons.
- **demonstrate the presence of extensive contralateral synaptic connectivity between clock neurons** that suggests a mechanism for clock synchronization across hemispheres to enhance robustness.
- highlight that light input to the circadian clock is largely indirect. Further, we **characterize these novel light input pathways to the clock neurons**, offering insight into the entrainment of the circadian clock to external light-dark cycles.
- **comprehensively elucidated output pathways from the clock network** to regions in the brain important for learning and memory, navigation, motor control, and hormone production and release. Surprisingly, our analyses uncovered infrequent monosynaptic connectivity between clock neurons and downstream output centers.
- use the connectome to generate specific hypotheses on how **daily rhythmic behaviors (i.e., feeding and reproductive behaviors) are regulated**.
- integrated single-cell transcriptomic analysis and receptor mapping to also **generate a putative peptidergic connectome of the clock network**. Our findings highlight the pivotal role of peptidergic signaling in enriching connectivity within the network and regulating rhythmic hormonal signaling.
- discuss broad **similarities and differences between *Drosophila* and vertebrate clock networks**.

Our findings significantly contribute to the field of circadian biology and neuroendocrinology. Given the high conservation of neuronal network motifs between *Drosophila* and vertebrates, this research will improve our understanding of circadian regulation in more complex organisms, including humans. Moreover, the circadian clock connectome serves as an excellent resource to systematically investigate circadian dysregulation which is linked to various health issues in humans including sleep, metabolic, and mood disorders. Thus, the findings presented in this manuscript will be of interest to the general public and the broader scientific community.

Response to reviewers

We would first like to thank the anonymous reviewers for their detailed comments on the earlier versions of our manuscript and for providing suggestions for further improvements. Below, we provide detailed responses (in red) to the comments from the two reviewers.

Referee #1 (Comments for the Author):

The author's response to two rounds of critiques is to respectfully disagree with my concerns. I think the authors have two options. 1.) If the work is to appear in a journal like Nature Communications, they must follow-up the new anatomical features they have characterized with experimental work to support the conclusions they make and remove the highly speculative sections of the study, particularly the speculations regarding non-synaptic peptidergic connectivity, which amount to reckless speculation presented as results. 2.) They could simply describe the interesting anatomical insights that their new connectomic analysis has provided, avoid the over interpretation of these results, discuss the possible implications of the new connections they have discovered, and publish the paper in a journal more appropriate for a work of neuroanatomy. Option one would require years of work. Option 2 would be a welcome and well cited addition to the literature that would be the basis for a great deal of future work. There are excellent journals that would be appropriate for such a study.

While we acknowledge this reviewer's frustration, we have tried to address several concerns from all reviewers over the two rounds of revision. Compared to the original submission, we have now:

- performed a detailed anatomical analysis of DN3 clock neurons by quantifying their numbers and classifying them in two morphologically-distinct categories (Fig 2A-I).
- characterized the morphology of all DN1p using multi-color flip out analysis (Fig 2J-N).
- included a more detailed analysis of individual inputs to and outputs from clock neurons (Fig 4D, Fig S13 and Fig S14).
- identified output pathways and generated specific testable hypotheses via which the clock could regulate rhythmic feeding and reproductive behaviors (Fig 5G-K).
- Identified fast-acting neurotransmitters expressed in clock neurons, thus providing valence to the output pathways (Fig 5D-K).
- substantially keyed down our conclusions on the paracrine connectivity within the clock network by modifying the title and text, and also performing more stringent analyses (Fig S18).
- restructured the manuscript by removing the characterization of endocrine cells as suggested by this reviewer.

The work presented in this manuscript therefore goes beyond a simple neuroanatomical analysis of clock neurons. Functionally characterizing the pathways presented here are simply beyond the scope of this study which, as acknowledged by this reviewer, would take years to perform. Nonetheless, we have now further modified the text where appropriate in order to reduce the speculations/hypotheses to avoid over interpretation of the results.

Below, I respond to the each of major points made by the authors, which are placed in quotations below.

"Accurate quantification of the total number of DN3 clock neurons for the first time which resulted in the identification of 90 additional clock neurons than previously estimated (Figure 2A-I). These additional novel neurons provide the major output to descending neurons that regulate behaviors, thus providing novel insights into how rhythmic behaviors are generated (Figure 4B and new figure 5)."

I concede that the authors revision of the total number of clock neurons is important and significant. But this observation is immediately followed by pure speculation that "these additional novel neurons provide the major output to descending neurons that regulate behaviors, thus providing novel insights into how rhythmic behaviors are generated." Absent experiments that support this conclusion, this is simply speculation presented as confirmed results.

We apologize if our rebuttal letter was not drafted properly and did not include the exact excerpts from the text. We would like to clarify that we do not use the same tone in the text as we did in rebuttal letter. We fully agree with the reviewer that the pathways identified using the connectome need to be experimentally validated and have made our speculations explicit. Below is an excerpt from the text relevant to this comment:

"Three descending neuron types, namely DNpe048, DNpe033 and DNpe041, receive the majority of clock inputs. Each of these descending neurons receive a substantial portion (about 25%) of their total input from clock neurons (Figure 5H). We focussed further on DNpe048 which receives strong cholinergic and glutamatergic inputs from clock neurons. DNpe048 express crustacean cardioactive peptide (CCAP) and have recently been shown to regulate sugar and water ingestion by integrating inputs from other modulatory interneurons (Gonzalez Segarra et al., 2023) (Figure 5I). Allatostatin-A-expressing Janu-AstA neurons which regulate hunger and thirst (Landayan et al., 2021) also provide inputs to DNpe048. It is thus tempting to speculate that DNpe048 integrate time and temperature cues via s-CPDN₃A-D and LPN along with hunger and thirst signals from diverse modulatory interneurons to regulate daily feeding rhythms (Figure 5I-J)."

We feel that this amount of speculation should we allowed, especially since our goal is to stimulate additional studies based on the connectivity presented in our study.

"We identified for the first time the morphology and connections of all DN1p (including the Cryptochrome-negative ones)

and classified them into 5 subgroups (the CRY-positive DN1pA and B, and the CRY-negative DN1pC-E). So far, only the function of some CRY-positive DN1p is known. The function of the other DN1p is unknown but according to our analyses, they are a major contributor to the clock output, and are predicted to play diverse roles based on their morphology and connectivity.”

While it is of interest to see the connectomic analysis extended to new subclasses of clock neuron, the authors once again resort to pure speculation to make the anatomical details take on a significance that is not yet established, going so far as to conclude that the DN1ps “...are a major contributor to the clock output.” Given that DN1ps are dispensable for most aspects of circadian timekeeping and sleep regulation, it's not clear how the authors can conclude this based on the new connections they have characterized. This has been my concern from the beginning with this study. Interesting new anatomical details are being used to make conclusions that would require a great deal of work to confirm.

It is true that the DN1ps are dispensable for circadian rhythmicity under constant conditions as shown in several studies (e.g. Nettnin et al. iScience 2021; Saurabh et al., J Biol Rhythms, 2024), including the study mentioned below with *glass* mutants lacking the DN1ps (Helfrich-Förster et al., Neuron 2001). However, in nature, organisms are not exposed to constant darkness, but to the light-dark and temperature cycles of the environment, and all studies conducted so far have shown that the DN1ps are important clock output neurons that fine-tune the phase of behavioral rhythms (Cavanaugh et al., Cell, 2014; Guo et al., Nature, 2016; Neuron, 2018; Chatterjee et al., Curr Biol, 2018; Yadlapalli et al., Nature, 2018; Goda et al., Sci Rep, 2019; Ni et al., eLife, 2019; Lamaze and Stanewsky, Front Physiol, 2020). In other words, the DN1ps are important for the correct timing of activity, sleep, feeding and other rhythms, and this is essential for survival. As we show here, the DN1ps are a very heterogeneous population of clock neurons that may even play opposing roles in the timing of activity, as we have recently found (Sekiguchi et al., J Biol Rhythms, 2024). While the presence of the clock protein PER in 4-6 CRY-positive DN1ps per hemisphere (in a *per⁰* background) was sufficient to generate robust morning and evening activity, the presence of PER in all 14 DN1ps (*per* rescue with the most used *Gal4* driver *Clk4.1M*) was less efficient. Most importantly, the role of the CRY-negative DN1ps is completely unknown, as all *Gal4* lines used so far either target all DN1ps or only the CRY-positive DN1ps. As we show here, CRY-positive DN1pAs mainly establish contralateral connections to other clock neurons, suggesting that they coordinate circadian oscillations between clock neurons of both hemispheres. In contrast, most CRY-negative DN1ps remain ipsilateral and have many output synapses exclusively to central brain neurons not part of the clock network, suggesting that they make an important contribution to clock output. This hypothesis can now be tested, and we will certainly do so. As the reviewer acknowledges, this test will require a lot of work that is beyond the scope of this paper.

Having said that, below is the only sentence from the results section regarding DN1p synaptic connectivity.

“Further, examination of individual postsynaptic partners of clock neurons (Figure S14) reveals that the majority of the output from DN1pA is onto other clock neurons, namely LNITP, LN_dCry+, and s-CPDN3C-D (Figure 4B and 4D). s-CPDN3C-D in turn provide strong output to DN1pC-E (Figure 4D-E), thus linking Cry-negative DN1p clusters to Cry-positive DN1p clusters.”

This is the only speculation we make regarding DN1p in the discussion:

“The previous analysis was based on only 24 clock neurons and largely focused on LN clusters, while excluding l-LNV, and several DN1p, DN2, and DN3 clusters due to the incomplete nature of the dataset. However, as evident from our analysis here, the DN in fact represent an important hub in the clock network and display high synaptic connectivity. In particular, DN1p play a large role in clock cluster interconnectivity and DN3 appear important for clock output pathways.”

Based on this, we do not think we have overinterpreted the results.

“While it was known that DN1pA project contralaterally, their downstream synaptic partners were only partially known. Our analysis convincingly shows that these contralateral projections are nearly exclusively to other clock neurons which suggests a mechanism for clock synchronization across the two hemispheres (Figure 1E-F). This is in contrast to other clock neurons (e.g., LN_d and DN3) which also project contralaterally but do not provide extensive inputs to other clock neurons. Hence, morphological and connectivity analyses do not always correlate.”

I agree that DN1 connectivity was only “partially known.” This was true for all neuronal classes and is still true to some extent: as comprehensive as this study attempts to be, new details will surely emerge as the clock network continues to be examined. Furthermore, I agree that the differential patterns of connectivity displayed by the various classes of clock neuron are of interest. But this analysis simply extends the work previously done, including a substantial amount of work done by the authors themselves. The idea that the pattern of contralateral connections formed by the DN1pAs “suggests a mechanism for clock synchronization across the two hemispheres” is complete speculation. Given that *glass* loss of function mutants, which lack DN1ps entirely appear to have no problem creating coordinated rhythms would appear to be strong evidence against this hypothesis, as they provided no behavioral evidence that the two hemispheres are not synchronized.

We respectfully disagree with the reviewer. Although *glass* mutants remain rhythmic under constant conditions, they have rather sloppy rhythms (see Figure 1 in Helfrich-Förster et al., Neuron 2001). They show a larger variance in the phases of morning activity between individual flies, they extend their evening and morning activity into the dark phase of the light-dark cycle, and under constant conditions both activity bouts are not tightly coupled. Thus, these mutants have problems in showing coordinated rhythms, and this is likely due to a failure in coordinating the rhythms of the two hemispheres. We

will test our hypothesis in future studies, but this clearly goes beyond the scope of the present paper, since it will take at least one year to carry out the relevant experiments.

Figure 1 from Helfrich-Förster et al. (2001)

Due to space limitations, we do not include these details in the discussion.

Below is the only excerpt in the discussion where we discuss synchronization of the two hemispheres:

“Analysis of interconnectivity within the clock network revealed extensive contralateral synaptic connectivity between the clock neurons, which is largely mediated by DN_{1p}A and to a lesser extent by s-CPDN₃C, s-CPDN₃D, l-CPDN₃, and LN^{ITP}. Furthermore, paracrine peptidergic signaling amongst clock neurons has the potential to further strengthen this contralateral connectivity. Such a strong bilateral coupling of clock neurons could prevent the internal desynchronization of clock neuron oscillations between the two hemispheres – a phenomenon that can happen in other insects and even in mammals, but so far has not been observed in fruit flies (Helfrich-Förster, 2004).”

We have now added “could” to tone down this statement.

“One of the major partners of the DN_{1p}A are the here newly described small central projecting DN₃ (s-CPDN₃ C–D). The s-CPDN₃ C–D project on the CRY-negative DN_{1p}C–E, which are main output neurons of the clock network and synapse on different central brain neurons. The function of these central brain neurons is still unknown but several of them make connections in the subesophageal ganglion and the superior medial protocerebrum and are suited to modulate these brain regions in a circadian manner.”

Again, this is interesting, and it provides some new insight into the synaptic connectivity underlying the clock neuron network. But the authors are, once again, taking a bit of new information and then speculating about the possible functional implications of these connections. This is fine of course, but I fear that the study repeatedly attempts, in describing the results, to make claims that it can't make without additional mechanistic work. Anatomical studies are valuable to the field and can be presented on their own, with along with some speculation about the functional import of the new anatomical features revealed being presented more cautiously in the discussion. It is important here to note that clock neurons have been shown to have non-circadian functions (e.g., PMID 22561453), so output pathways from clock neurons may underly non-circadian functions.

We would again like to clarify that the quote above from the rebuttal letter is not directly adopted from the text. Below is the exact quote from the text:

“Further, examination of individual postsynaptic partners of clock neurons (Figure S14) reveals that the majority of the output from DN_{1p}A is onto other clock neurons, namely LN^{ITP}, LN^{dCry+}, and s-CPDN₃C–D (Figure 4B and 4D). s-CPDN₃C–D in turn provide strong output to DN_{1p}C–E (Figure 4D–E), thus linking Cry-negative DN_{1p} clusters to Cry-positive DN_{1p} clusters.”

The reviewer is correct in their view that not all timing is circadian. There are also rhythms on other time scales such as

ultradian courtship song rhythms (Kyriacou and Hall, PNAS 1980) and infradian seasonal rhythms (e.g. Nagy et al. PLoS Genetics 2019; Hildalgo et al. Curr Biol 2023) that depend on specific circadian clock genes. In addition, there are other temporal phenomena, such as the copulatory timing mentioned here, which also depend on the clock (Beaver and Giebultowicz, Curr Biol 2004; Kim et al. Nature Neurosci 2012). Since all these phenomena are related to timekeeping, it seems plausible that circadian clock genes and clock neurons are involved. We have not mentioned these phenomena in the discussion for reasons of space but can certainly do so if necessary. It is interesting to note that copulatory timing involves memory processes and that these are not carried out via the mushroom bodies but via the ellipsoid body (Kim et al. Nature Neurosci 2012). In our study, we found only sparse synaptic connections between the clock neurons and the ellipsoid body, but the PDF receptor is expressed in the ellipsoid body (Pérez et al., J Neurophysiol 2013), implying that signaling from the LNV to the ellipsoid body neurons may occur via the neuropeptide PDF. However, the inclusion of this putative signaling pathway in our manuscript would be highly speculative, much more so than the other conjectures we have made. For this reason, we have refrained from doing so.

“We have also performed comprehensive trans-Tango-based circuit tracing analyses within the clock network to complement the connectome analyses (Figure S9-10).”

This is interesting, but, in addition to the synapses detected in the adult connectome, trans-Tango will reflect synapses that form during development and subsequently disappear along with synapses that are present during the first days of adulthood that no longer exist in the mature adult. So, in my opinion, the authors are likely obscuring the clear picture provided by the new connectomic analysis with the use of this technique.

We agree with the reviewer that synapses formed during development will no longer exist in mature adults. That is precisely the reason why we used 2-week old adults for our trans-Tango analyses to avoid/reduce developmental influence on adult connectivity. Further, our trans-Tango analysis for most neurons agrees with our connectome analyses quite well so we don't see how it is obscuring the picture.

We forgot to include the age of the animals in the methods. This has now been clarified.

“We have also discovered additional neuropeptides expressed in the clock neurons using two different approaches (single-cell RNA sequencing analysis and immunohistochemistry) (Figure 6). The functions of these neuropeptides within the clock network will be characterized in a future study.”

Again, an excellent start to what is likely to be an insightful line of questioning. An observation warranting description in a high quality journal that features neuroanatomical studies. Interesting to the field, but we must await mechanistic experiments to reveal what, if anything, these peptides do.

Mechanistic studies on novel clock neuropeptides identified using expression data are routinely published as stand-alone studies in top-tier journals in the field. For example, the discovery and function of Ast-C by Diaz et al (<https://doi.org/10.1016/j.cub.2018.11.005>) was published in Current Biology, and the role of ITP in the clock system by Hermann-Luibl et al (<https://doi.org/10.1523/JNEUROSCI.0111-14.2014>) was published in Journal of Neuroscience. Undertaking mechanistic experiments to this degree are thus beyond the scope of this study.

“Additional analysis to show that s-CPDN3 neurons function as integrators by linking different subclasses of DN1ps (new Figure 4E). This is important because the newly described CRY-negative DN1pC-E have outputs to still unknown central brain neurons that may be involved in the transfer of clock signals to other brain regions and beyond.”

This is interesting, and I agree that the DN3 work is the most novel aspect of the work.

We are glad that the reviewer appreciates this aspect of our study.

“Fast-acting neurotransmitter expressed in all the clock neurons based on EM-based neurotransmitter prediction as well as immunohistochemical analyses. We show that glutamate and acetylcholine, but not GABA are the major neurotransmitters in clock neurons (new Figure 5D-F). The inclusion of valence to the synaptic connectivity will facilitate future studies modeling clock-driven behaviors.”

This is also speculation. The EM predictions and GAL4 expression supporting these conclusions amount to an intriguing set of observations that I hope will be followed up with functional studies. Absent follow-up studies, these observations remain speculative and are better suited for a journal that features anatomical work.

We are not sure how our analyses can be classified as speculative. Acetylcholine is universally regarded as an excitatory neurotransmitter and GABA as inhibitory. We acknowledge that glutamate can either be excitatory or inhibitory and have mentioned this in the text. However, our mapping of neurotransmitters across the clock network allows future studies to specifically interfere with the appropriate neurotransmitters to determine their function instead of screening for multiple neurotransmitters within a given cell population. Hence, our analysis is yet another step towards deciphering functional connectivity within the clock network.

“A more detailed analysis of how the circadian system is connected to descending neurons. Specifically, we show that s-CPDN3 and LPN provide major direct outputs to descending neurons (new Figure 5G).”

Here the authors offer more details on output pathways from clock neurons. This is of interest and may end up being the basis of future experimental work, but does not represent major new findings that move the field forward.

We thank the reviewer for appreciating our new analysis.

“We postulate pathways via which the clock regulates rhythmic feeding and reproductive behaviors to highlight the utility of this resource in generating novel hypotheses that can be experimentally tested (new Figure 5H-K).”

Such postulation is a welcome part of the discussion of the potential functions of the output pathways characterized, but this does not amount to a “novel insight,” as stated by the authors.

Since the hypotheses/pathways presented in Figure 5H-K have not been reported previously, we don't think it is an exaggeration to refer to them as novel. We do agree that these mechanisms are putative and deserve to be functionally tested and have mentioned this in the text under limitations in the discussion.

“The novel synaptic and putative paracrine connections reported here can only be considered predictions until they are functionally verified.”

“Lastly, we agree with this reviewer that the connectome of Pars Intercerebralis would be of great interest and deserves to be pursued in a greater depth. Since reviewer 2 also suggested to reorganize the manuscript and make it less overwhelming, we have now removed our identification of neuroendocrine cells from the manuscript. This allows us to focus more on clock neurons (modified Figure 4 and new Figure 5).”

I am happy to see that the authors agree with something that I've expressed.

We thank the reviewer for prompting us to restructure this part of our study. We have now followed their suggestion and performed an in-depth analysis of endocrine cells: <https://www.biorxiv.org/content/10.1101/2024.08.28.609616v1> .

“It is not clear to the authors which previous connectomic studies have shown light and other sensory inputs to clock neurons. The hemibrain data did not include the optic lobe and the ocelli. Similarly, putative inputs from other sensory modalities (e.g., temperature) were also not examined in previous connectomic studies. Hence, our study is the first to highlight all the pathways from sensory neurons to clock neurons.”

Work from the Jefferis lab has described thermal inputs onto the DN1as and DN1ps (PMID: 32619476), work in the Fernandez lab has characterized inputs onto the lateral neuron classes (PMID: 35766361), including inputs from the HB eyelet and accessory medulla. The authors themselves has examined inputs, including temperature inputs onto the DN1ps (PMID: 35574472) and have described synaptic inputs onto the LPNs (PMID: 34961936).

We apologize for the sloppiness in our response in the rebuttal letter. In fact, the Jefferis lab (Marin et al., 2020) has conducted a connectome analysis of the whole adult female brain (FAFB) to uncover the connectivity of the antennal glomeruli involved in temperature and humidity sensing. This study identified indirect thermosensory input pathways to clock neurons, which was the reason we focused mainly on direct thermosensory sensory input to clock neurons. We have now added a short sentence (in italics) to the manuscript clarifying this: “This suggests that the clock receives strong light inputs from extrinsic photoreceptor cells, albeit indirectly. *Similarly, thermosensory input to clock neurons is also indirect via thermosensory projection neurons (Marin et al., 2020).*”

Further, we have cited a personal communication from Gregory Jefferis confirming that the only four sensory neurons that provide direct input to clock neurons are thermosensory neurons. Nevertheless, the other three studies, including our own (Shafer et al., 2022; Reinhard et al., 2022a,b) have used the hemibrain that lacks the optic lobes and ocelli. Therefore, the present study is actually the first to show all pathways from the photoreceptor cells to the clock neurons. The surprising and novel finding of the present study is that most of the photoreceptor input to the clock neurons is indirect. This does not exclude the direct input from the HB eye cells to the clock neurons described by us and others. However, this direct input is very weak compared to the massive indirect input via accessory medulla neurons. Therefore, we consider it extremely important to describe these indirect connections in detail.

“The only similarity between our study and the two companion connectome papers in the FlyWire package is that they looked at ocelli pathways. Our analyses complement their analyses and go beyond as we also focus on light input from structures other than the ocelli. The companion paper only examined the connectivity from the ocelli to ocellar ganglion cells. This is in contrast to our analysis where we show how the light input via the ocelli could travel to the clock neurons. Hence, we don't see any redundancy in our analysis and the other connectome studies. Nonetheless, since we are using the connectome and broad annotations described in the two companion connectome papers, we feel that it is appropriate to cite them where relevant.”

I agree that the inclusion of the ocelli is a most welcome addition to the literature and a source of novelty in the present study.

We are glad that the reviewer appreciates this aspect of our study.

“We agree with the reviewer that we, as a scientific community, do not know much about how far peptides can diffuse through the brain. We have also acknowledged this issue in the manuscript. Nonetheless, our analyses used to determine putative peptidergic connections are a step above previous studies utilizing just expression data (see: Figure 2 Supplement 4 in <https://elifesciences.org/articles/65745>). We showed putative peptidergic connectivity under 2 situations....We think that the actual amount of paracrine signaling is somewhere in between the two extremes and our threshold of 14 micro diffusion distance seems a good starting point to examine this mode of signaling.”

I remain deeply skeptical of this aspect of the study. We simply do not understand non-synaptic peptidergic signaling in the intact brain well enough to present anything like the “putative peptidergic connectivity” presented in this paper. This is a central question of great importance that will require a great deal of work examining the physiology of such connections and the anatomical rules that govern peptide diffusion through the brain. This aspect of the work amounts to reckless speculation in my opinion and should not be published without functional studies, even with the “third [more conservative] situation” added by the authors.

We have made the analysis on putative peptidergic connections more stringent over the two rounds of revision in line with a previous study in Neuron (see: <https://doi.org/10.1016/j.neuron.2023.09.043>). Other studies have also used gene expression data to predict connectivity between neurons (see: Figure 2 Supplement 4 in <https://elifesciences.org/articles/65745> and Figure 3 in <https://elifesciences.org/articles/26349>). Our analysis combines the gene expression data with the constraints determined by the connectome (see methods and Fig S18 for the detailed approach). This is more stringent than using just the gene expression data. Our main conclusion from this analysis is that paracrine signaling could enrich the connectivity between clock neurons and we do not over speculate on any specific peptidergic pathways. We have also listed all the limitations of our approach to avoid false interpretations of our analysis. However, if other reviewers and the editor share the opinion of reviewer 1, we are open to removing Fig 7E and Fig S18 from the manuscript.

In conclusion, I think there is much of importance and interest in the new anatomical features described in this study, but there is simply too much speculation and over-interpretation, and this remains a serious weakness of the study.

As mentioned above for some of the points, our rebuttal letter included points which were rephrased from the manuscript. If there are specific examples of over speculation or over interpretation within the manuscript itself, we are happy to modify it further.

Reviewer #2 (Remarks to the Author):

The revised manuscript by Reinhard, Fukuda, and colleagues has significantly improved compared to their original submission to Nature, after multiple rounds of revisions. This study has made several major findings that will be of great interest to the fly circadian field:

1. This study expanded the number of clock neurons from approximately 150 to 242 by adding and characterizing new members of the mysterious DN3 and DN1 subgroups.
2. The research mapped out the connectivity within the circadian neural network and detailed the input/output profiles of circadian neurons.
3. Notably, the additional analysis presented in this version (Figure 4 DE and Figure 5) provides several clear roadmaps for future research on the output pathways of circadian regulation of various behaviors, such as feeding, drinking, and mating.
4. By combining the EM prediction dataset (Eckstein et al., 2024) and the scRNAseq dataset (Ma et al., 2021), the study has assigned neurochemical labels (molecular basis) to the connectivity they mapped.

Together, these findings assemble a comprehensive circadian clock connectome. This will become an important resource for the fly circadian field. I did not find any other major concerns and look forward to exploring the dataset.

We thank this reviewer for their constructive feedback throughout the process which has considerably improved the manuscript. We are glad for their endorsement and agree that our study will be an important resource for the circadian community.

Response to reviewer

We would first like to thank the anonymous reviewer for their comment on our manuscript and our response to Reviewer 1's comments. We have revised the manuscript based on their suggestions.

Reviewer comments:

As requested, I'm commenting mostly on the authors' response to Reviewer 1. Overall, the authors have addressed these comments (see exceptions below). And I believe they are justified in not doing additional experiments to support their predictions. Some amount of speculation is acceptable although this should be of a general nature and refrain from specific claims, which can only come from experimental evidence.

We thank the reviewer for validating our justifications.

I note though that the author response to the reviewer's comment about trans-Tango is not correct. The reviewer noted that trans-Tango can be misleading as it may report synapses that occurred earlier in life. In response, the authors said that they used 2 week old adults for trans-Tango. However, as long as the relevant transgenes were expressed all through development (which appears to be the case), they may still be detecting synapses that occurred at earlier stages of life. This should be acknowledged. It may even explain why some of their trans-tango experiments detected more synapses than indicated by the connectome.

We acknowledge this concern and have now included it in the discussion that follow these results. The modified sentence reads as follows:

"This discrepancy could be explained by: 1) the presence of additional neurons in the Gal4-line (e.g. PDF tritocerebrum neurons, 2) daily remodeling of neural circuits, as shown previously for s-LN_v and DN_{1a} and/or 3) connections persisting through development."

Also, I was confused by the following response regarding the authors' efforts to trace sensory projections to clock cells: *"We apologize for the sloppiness in our response in the rebuttal letter. In fact, the Jefferis lab (Marin et al., 2020) has conducted a connectome analysis of the whole adult female brain (FAFB) to uncover the connectivity of the antennal glomeruli involved in temperature and humidity sensing. This study identified indirect thermosensory input pathways to clock neurons, which was the reason we focused mainly on direct thermosensory sensory input to clock neurons. We have now added a short sentence (in italics) to the manuscript clarifying this: "This suggests that the clock receives strong light inputs from extrinsic photoreceptor cells, albeit indirectly. Similarly, thermosensory input to clock neurons is also indirect via thermosensory projection neurons (Marin et al., 2020)."*

Further, we have cited a personal communication from Gregory Jefferis confirming that the only four sensory neurons that provide direct input to clock neurons are thermosensory neurons. Nevertheless, the other three studies, including our own (Shafer et al., 2022; Reinhard et al., 2022a,b) have used the hemibrain that lacks the optic lobes and ocelli. Therefore, the present study is actually the first to show all pathways from the photoreceptor cells to the clock neurons. The surprising and novel finding of the present study is that most of the photoreceptor input to the clock neurons is indirect. This does not exclude the direct input from the HB eye cells to the clock neurons described by us and others. However, this direct input is very weak compared to the massive indirect input via accessory medulla neurons. Therefore, we consider it extremely important to describe these indirect connections in detail."

It seems that in the first para they are saying thermosensory input is indirect, and in the second that Greg Jefferis has shown direct input. Also, if this was already done, then why does the first para say "we focused mainly on direct thermosensory sensory input to clock neurons"? The manuscript also did not clarify this issue.

We apologize for the confusion. The direct input from four thermosensory neurons is rather weak and most of the strong thermosensory input to clock neurons is indirect. We have now modified the text as follows:

"Interestingly, only 4 sensory neurons provide direct inputs to the clock network. These are anterior cells (aDT4) (Dr. Gregory Jefferis, personal communication) (Figure 3A and D) which provide temperature inputs to LPN, DN_{1pC}, and DN_{1pE} (Jin et al., 2021, Alpert et al., 2022). Therefore, most of the strong thermosensory input to clock neurons is indirect via thermosensory projection neurons (Marin et al., 2020)."